## TOOLS

# High-content imaging-based pooled CRISPR screens in mammalian cells

Xiaowei Yan[1], Nico Stuurman[1], Susana A. Ribeiro[1,2], Marvin E. Tanenbaum[1,3], Max A. Horlbeck[1,4], Christina R. Liem[1,5], Marco Jost[1], Jonathan S. Weissman[1,6], and Ronald D. Vale[1,7]

**CRISPR (clustered regularly interspaced short palindromic repeats)-based gene inactivation provides a powerful means for linking genes to particular cellular phenotypes. CRISPR-based screening typically uses large genomic pools of single guide RNAs (sgRNAs). However, this approach is limited to phenotypes that can be enriched by chemical selection or FACS sorting. Here, we developed a microscopy-based approach, which we name optical enrichment, to select cells displaying a particular CRISPR-induced phenotype by automated imaging-based computation, mark them by photoactivation of an expressed photoactivatable fluorescent protein, and then isolate the fluorescent cells using fluorescence-activated cell sorting (FACS). A plugin was developed for the open source software µManager to automate the phenotypic identification and photoactivation of cells, allowing ~1.5 million individual cells to be screened in 8 h. We used this approach to screen 6,092 sgRNAs targeting 544 genes for their effects on nuclear size regulation and identified 14 bona fide hits. These results present a scalable approach to facilitate imaging-based pooled CRISPR screens.**

## Introduction

High-throughput sequencing in combination with CRISPR technology has greatly accelerated discoveries in biology through unbiased identification of many new molecular players in key biological processes (Hsu et al., 2014; Barrangou and Doudna, 2016; Kweon and Kim, 2018; Schuster et al., 2019). Using a high-diversity sgRNA library, large numbers of genes can be manipulated simultaneously in a pooled manner, and sgRNA abundance differences can be determined with high-throughput sequencing quickly, with low labor and financial cost. Thus far, pooled CRISPR screens have been limited to phenotypes that can be transformed into sgRNA abundance differences, such as growth effects (Gilbert et al., 2014; Shalem et al., 2014; Wang et al., 2014). or phenotypes that can be directly examined by flow cytometry (Parnas et al., 2015) or single cell molecular profiling (Dixit et al., 2016; Jaitin et al., 2016; Datlinger et al., 2017; Adamson et al., 2018 *Preprint*; Wroblewska et al., 2018; Rubin et al., 2019). However, many important cellular phenotypes can be detected only under a microscope, which requires a robust method for transforming optically identified phenotypes into differences in sgRNA abundance. Arrayed sgRNA libraries greatly facilitate such microscope-based screens, but are not widely available. Several in situ sequencing (Feldman et al., 2019; Wang et al., 2019) and cell isolation

(Chien et al., 2015; Piatkevich et al., 2018; Wheeler et al., 2020) methods have been developed that allow microscopes to be used for screening. However, these methods contain non–high-throughput steps that limit their scalability.

Recently, an imaging-based method named "visual cell sorting" was described that uses the photoconvertible fluorescent protein Dendra2 to enrich phenotypes optically, enabling pooled genetic screens and transcription profiling (Hasle et al., 2020). Here, we developed an analogous approach to execute an imaging-based pooled CRISPR screen using optical enrichment by automated photoactivation of the photoactivatable fluorescent protein, PA-mCherry. Similar to traditional enrichment-based pooled CRISPR screens, cells are infected with an sgRNA library, and high-throughput sequencing is used to examine sgRNA abundance. Instead of traditional enrichment strategies, we use optical enrichment: cells exhibiting the desired phenotype are identified and photoactivated automatically under a microscope. Photoactivated cells are then isolated using flow cytometry and analyzed by high-throughput sequencing. We first evaluated this approach using a synthetic fluorescent reporter to estimate screening accuracy and capacity. We then applied this approach to identify genes that regulate nuclear size. This methodology is modular, allows millions of cells to be

[1]Department of Cellular and Molecular Pharmacology, Howard Hughes Medical Institute, University of California, San Francisco, San Francisco, CA; [2]Cairn Biosciences, Inc., San Francisco, CA; [3]Oncode Institute, Hubrecht Institute-KNAW and University Medical Center Utrecht, Utrecht, Netherlands; [4]Boston Children's Hospital, Boston, MA; [5]University of California, San Diego, San Diego, CA; [6]Whitehead Institute and Department of Biology, MIT, Cambridge, MA; [7]Janelia Research Campus, Howard Hughes Medical Institute, Ashburn, VA.

Correspondence to Ronald D. Vale: valer@janelia.hhmi.org.

Figure 1. **Imaging-based pooled CRISPR screen.** Schematic of imaging-based pooled CRISPR screen. Cells expressing PA-mCherry are infected with a pooled sgRNA library and imaged using a microscope. Images are collected and analyzed automatically to generate an activation mask targeting cells of interest. Exposure with blue light photoactivates cells of interest into mCherry-positive cells that are subsequently isolated by FACS. Samples are analyzed by high-throughput sequencing for sgRNA identification.

screened within a few hours, and can be scaled to a genome-wide level.

## Results

### An optical approach for cell enrichment by patterned illumination followed by FACS sorting

We developed an approach, which we term optical enrichment, to select cells of interest using a microscope and mark them by photoactivation, enabling cell isolation using FACS (Fig. 1). To achieve this, we engineered hTERT-RPE1 cells expressing the photoactivatable fluorescent protein PA-mCherry and observed them under a microscope. A photoactivatable fluorescent protein was chosen over a photoconvertible fluorescent protein to increase the number of channels available for imaging. PA-mCherry was chosen to leave the better-performing green channel open for labeling of other cellular features. Moreover, nonactivated PA-mCherry has low background fluorescence in the mCherry channel (Fig. S1 b), and it can be activated to different intensities when photoactivated for various amounts of time. Cells of interest were selected by automated image analysis and then photoactivated with patterned illumination using a

digital micromirror device (DMD; Fig. S1 a). To avoid undesired photoactivation of neighboring cells, we limited the activation pattern to nuclei as identified by the H2B-mGFP signal (Fig. S1 b). We developed a plugin for the open-source microscope control software μManager (Edelstein et al., 2014) called Auto-PhotoConverter that automates these steps and has a pluggable interface for image analysis code so that it can be used for any desired phenotype (https://github.com/nicost/mnfinder; Fig. S1 c). After harvesting the cells, the photoactivated cells were isolated by FACS. By varying the activation time of the PA-mCherry, we were also able to create multiple populations of cells with different intensities that were distinguishable by FACS (Fig. 2, a and b), enabling analysis of multiple phenotypes simultaneously, as discussed below.

We next tested the precision of our automated photoactivation platform in a "mock screen" consisting of a mixture of cells expressing the fluorescent marker monomeric infrared fluorescent protein (mIFP) and cells not expressing mIFP (outlined in Fig. 2 c). In this mock screen, mIFP fluorescence was used as a "phenotype" to indicate cells of interest (mIFP-positive cells). The Auto-PhotoConverter plugin was used to identify and generate an activation mask based on mIFP fluorescence to

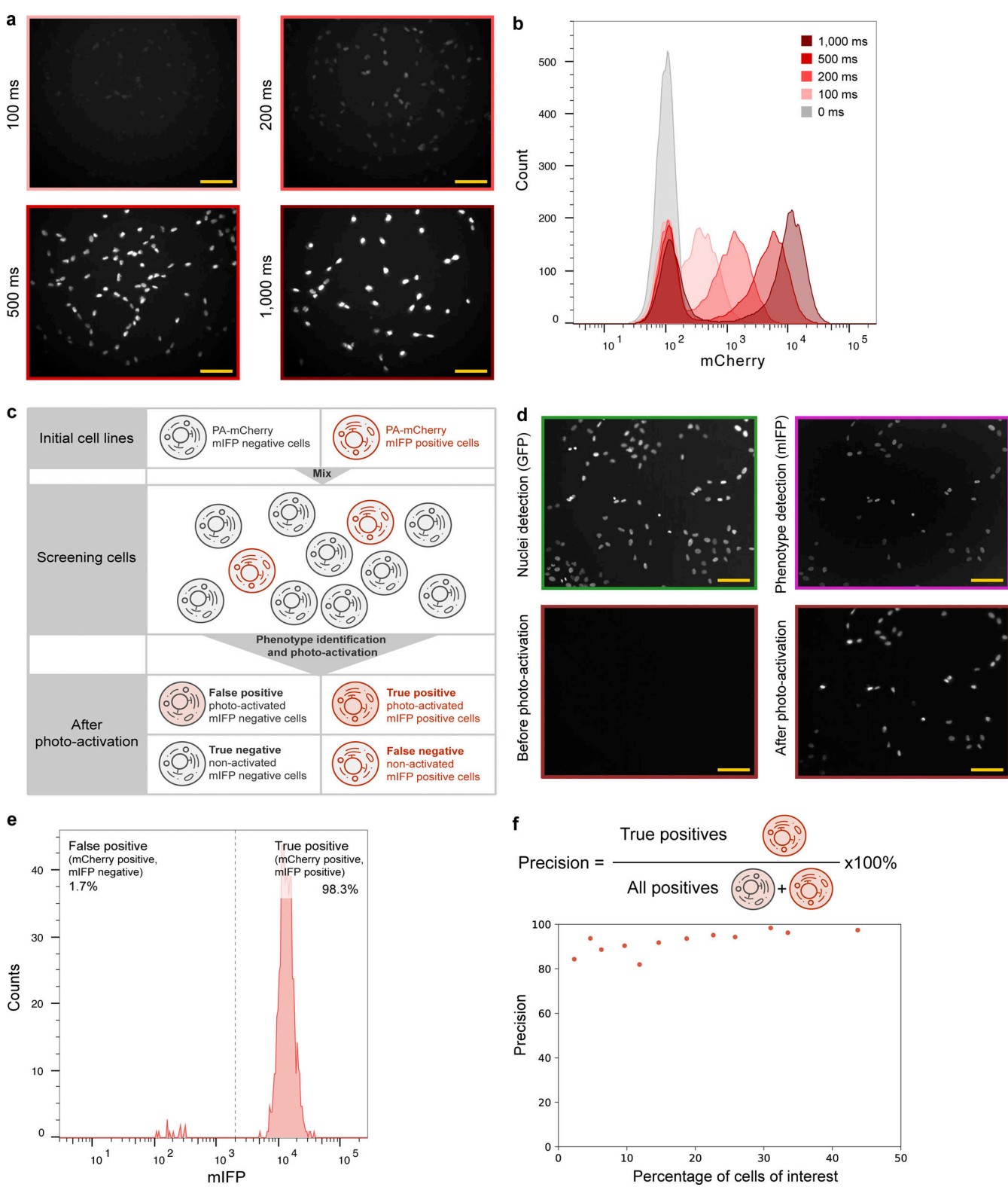

Figure 2. **Optical enrichment enables accurate cell identification and isolation. (a and b)** Cells can be activated into different fluorescent intensity levels that are clearly distinguished by FACS. Example images of cells (hTERT-RPE1 PA-mCherry) that have undergone various photoactivation times (a) and their corresponding FACS results (b; number of cells analyzed for each condition is ~10,000, replicate number = 2). Scale bar: 100 μm. **(c)** Schematic of the experiment to measure precision. mIFP-positive (hTERT-RPE1 PA-mCherry H2B-mGFP mIFP-NLS) and mIFP-negative cells (hTERT-RPE1 PA-mCherry H2B-mGFP) were mixed and analyzed under GFP and mIFP channels separately. mIFP expression was used as a phenotype to indicate cells of interest (mIFP-positive cells). An activation mask was generated for each mIFP-positive cell. Cells identified by FACS to be mIFP and mCherry double-positive are true positives, while mCherry-positive cells without mIFP fluorescence result from accidental activation (false-positive cells). **(d and e)** Cells of interest can be identified by automated image analysis followed by photoactivation and distinguished through FACS with high accuracy. **(d)** Example images of image analysis (GFP channel,

green; mIFP channel, pink), before and after photoactivation (mCherry channel, red) are shown. Scale bar: 100 μm. **(e)** Example FACS data (number of cells analyzed = 662). **(f)** Different ratios of mIFP-positive cells and mIFP-negative cells were mixed to measure precision at different percentages of hits.

photoactivate the mIFP-expressing cells into mIFP-mCherry double-positive cells (true-positive cells). Cells without mIFP fluorescence might also be accidentally photoactivated, leading to false-positive cells (mCherry single-positive cells). To evaluate the precision (the fraction of called positives that are true positives) of this assay, all cells were collected and analyzed by FACS after image analysis and photoactivation (Fig. 2, d and e). We calculated precision as the fraction of photoactivated cells (mCherry-positive cells) that were true positives (mIFP-mCherry double-positive cells; Fig. 2 f). When the initial subset of mIFP-positive cells was 30%, the precision was 98.3% (Fig. 2 e). The precision varied with the initial percentage of mIFP-positive cells and ranged from 80% to ~100% (initial percentage of mIFP-positive cells ranging between 2.3% and 43.7%; Fig. 2 f). Precision is expected to fall <80% with an initial percentage of mIFP positive cells <2.3%. However, these results indicate that optical enrichment can be used to identify hits with high precision even at relatively low hit rates.

**Optical enrichment enables accurate sgRNA identification**

Having established that we can recover photoactivated cells with high precision, we next tested if we can successfully identify specific sgRNA sequences present in these cells. mIFP-negative cells and mIFP-positive cells were separately infected with two different CRISPRa sgRNA libraries (6,100 sgRNAs for mIFP-negative cells; 860 sgRNAs for mIFP-positive cells) at a low multiplicity of infection to guarantee a single sgRNA per cell. Note that in these experiments, the sgRNAs function only as barcodes to be read out by sequencing but do not cause phenotypic changes, as the cells do not express corresponding CRISPR reagents. These two populations were then mixed at a ratio of 9:1 mIFP-negative cells to mIFP-positive cells. We again used mIFP expression as our phenotype of interest (outlined in Fig. 3 a). Two biological replicates were performed, and 200-fold coverage of each sgRNA library was guaranteed throughout the screen, including library infection, puromycin selection, imaging/photoactivation, and FACS. For each replicate, we screened a single imaging plate. A total of 1,825,740 and 1,490,188 RPE-1 cells in the two replicates were imaged separately using a 20× objective. Automated imaging and photoactivation of the plate took ~8 h. The mCherry-positive cells were isolated by FACS, and cells passing through the sgRNA gate without further analysis were also collected as a control (Ctrl; unanalyzed sample; FACS gating strategies are detailed in Data S1). These cells were separately prepared for high-throughput sequencing for sgRNA information extraction.

For simplicity, we use the terms "mIFP sgRNAs" for the sgRNAs used to infect mIFP-positive cells and "Ctrl sgRNAs" for the sgRNAs used to infect mIFP-negative cells. Typically, sgRNA libraries contain multiple sgRNAs that target a single gene, which minimizes confounding effects that arise from differences in sgRNA efficacy. Because the mIFP positive phenotype is not induced by our sgRNA library, we emulated genes in normal

sgRNA libraries in our analysis by grouping different numbers of randomly selected sgRNAs.

Our results show that the sgRNA groups from mIFP-positive cells (mIFP groups) could be well distinguished from the sgRNA groups in mIFP-negative cells (Ctrl groups; Fig. 3 b). To further investigate how library composition and number of replicates influence screening results, we also analyzed the data by grouping the sgRNAs differently (either one or two sgRNAs were assigned to each group) and two different numbers of replicates (phenotypic scores calculated from one replicate versus phenotypic scores averaged between two replicates). As shown in Fig. S2, mIFP sgRNAs could be distinguished from Ctrl sgRNAs in a single experimental replicate (Fig. S2, top left). Combining data from both replicates significantly improved segregation of the mIFP and Ctrl groups (Fig. S2, top right). Not surprisingly, the greater the number of sgRNAs assigned to a group, the better the detection of mIFP groups (Fig. S2, bottom). Two sgRNAs per group is sufficient for a reliable screening result, even using a single replicate (Fig. S2 bottom left). Thus, we demonstrate that pooled CRISPR libraries can be screened for phenotypes under a microscope by optical enrichment.

**Improved phenotype identification through multi-intensity labeling**

In most pooled CRISPR screens, only cells showing the phenotype of interest are selected, and the relative enrichment of a given sgRNA is calculated based on comparison with the whole cell population. However, this whole cell population is usually collected separately and includes both positive and negative cells, which reduces the perceived enrichment in the positive population. We therefore investigated calculating the relative enrichment of a given sgRNA by comparing with true-negative cells. Not all mCherry-negative cells are true-negative cells, since there are unanalyzed regions outside of the microscope field of view (gray region in Fig. 3 c, top) and cells that fail to pass the filters for phenotype identification (Data S2). Thus, true-negative cells also need to be labeled before harvesting. This task requires selecting for multiple phenotypes simultaneously. We achieved this within the same experiment using different photoactivation times for true positives (2 s) and true negatives (100 ms) and separating them by FACS (Fig. 3 c). For comparison, we also collected cells going through the same experimental procedure that were not analyzed during image analysis (unanalyzed cells, mCherry-negative cells) to determine the sgRNA composition in the total cell population. As shown in Fig. 3 d, the peaks indicating groups of mIFP sgRNAs and Ctrl sgRNAs were separated to a much greater extent when comparing with true-negative cells rather than with the whole cell population (Fig. 3 b), demonstrating that this approach can indeed improve screening of pooled sgRNA libraries. Additionally, this approach can be used to screen for multiple different phenotypes simultaneously.

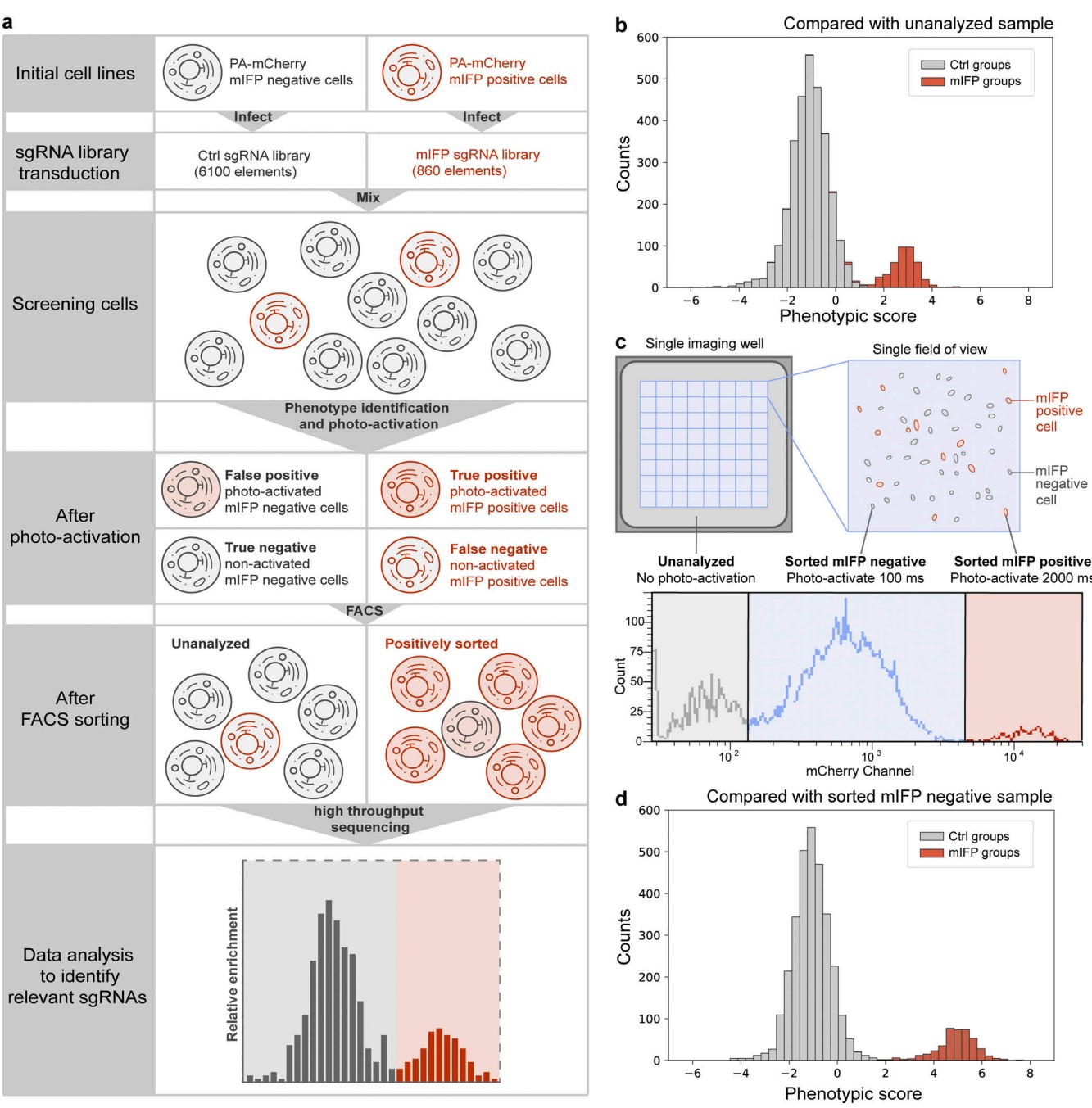

Figure 3. **Optical enrichment enables accurate sgRNA identification in a pooled CRISPR screen. (a)** Schematic of mIFP proof-of-principle screen. A mixed population of mIFP-positive and -negative cells was imaged and photoactivated as described in Fig. 2 c. mCherry-positive and unanalyzed cells were then isolated by FACS and prepared for high-throughput sequencing to extract sgRNA information. **(b)** Screening result presented by distribution of phenotypic scores of all the sgRNA groups. Red and gray, mIFP groups and Ctrl groups calculated by comparing with unanalyzed sample. **(c)** Schematic of dual-activation experiment. Experiment as described in Fig. 3 a, but mIFP-negative cells were also photoactivated (100 ms). mIFP-positive cells were activated using a longer activation time (2,000 ms) to guarantee a clear distinction by FACS. Image acquisition generally does not cover the complete imaging well, which leaves cells not imaged and unanalyzed. Lower panel shows an example of FACS data. Cells sorted for mIFP expression (sorted mIFP-positive), cells sorted for lack of mIFP (sorted mIFP-negative), and unanalyzed cells were separately collected and prepared for high-throughput sequencing. **(d)** Distribution of phenotypic scores of all the sgRNA groups compared with the sorted mIFP-negative sample. Phenotype identification is improved by comparing with true negative cells rather than unanalyzed cells as shown in Fig. 2 b. Red and gray, mIFP groups and Ctrl groups calculated by comparing with sorted mIFP negative sample.

## Pooled CRISPR screen for factors involved in nuclear size regulation

To further test our screening method, we performed a screen for regulators of nuclear size. We generated a CRISPRi library of

6,092 sgRNAs targeting 544 genes (10 sgRNAs/gene with 22 nontargeting sgRNAs) whose translation efficiency is up-regulated during the G2 phase of the cell cycle. This library includes sgRNAs targeting *FBXO5*, which is known to cause larger

nuclei after knockdown (Machida and Dutta, 2007; Verschuren et al., 2007) and served as the positive Ctrl. For this experiment, hTERT-RPE1 cells were further engineered with a CRISPRi modality (dCas9-KRAB-BFP) to inhibit transcription of genes targeted by the sgRNA library. We defined nuclear size as the 2D area in square micrometers measured by H2B-mGFP using an epifluorescence microscope, as determined by automated image analysis (Fig. 4 a and Data S2). We selected two Ctrl sgRNAs that have no targeting sites in the human genome and, as expected, had no discernible effect on nuclear size (Fig. S3 a). Nuclear sizes were measured for Ctrl cells, and the value of the top 0.5% was used as the screening threshold (1,000 μm²).

Positive cells were photoactivated and sorted together with unanalyzed cells as a comparison. Two biological replicates were performed containing 5,521,518 and 5,795,313 RPE-1 cells in total, each consisting of four imaging plates. Both replicates were completed within 2 d (each plate taking 7–10 h of imaging/photoactivation). The four imaging plates per replicate were performed as separate screening experiments, termed runs, and data were combined only after sgRNA abundance determination (Fig. 4 b). Simulated negative Ctrls were generated computationally by randomly regrouping all the sgRNAs (10 sgRNAs/group), and a phenotypic score was calculated for each gene and simulated negative Ctrl as described in Materials and methods. A score η summarizing effects from both severity of the phenotype (phenotypic score) as well as trustworthiness of the phenotype [−ln(P value)] were calculated, and an empirical false discovery rate (eFDR) = 0.1% was used to call hits for further analysis (Fig. 4 c and Fig. S3, b and c). The two replicates correlate well (Fig. 4 d) and combined yielded 30 hits, of which 15 genes were found in both replicates, including the positive Ctrl FBXO5 (Fig. 4 e).

To estimate the minimum requirements for performing an optical enrichment pooled CRISPR screen, we computationally analyzed the effect of both library composition and number of runs on the screening results. Using data from replicate 2, we reran the analysis, comparing results when fewer sgRNAs per gene and/or fewer runs were included. We binned the sgRNAs based on three commercially available CRISPRi libraries: 10 sgRNAs/gene and the "Top5" or "Supp5" subpools of the sgRNA library (Horlbeck et al., 2016). Top5 and Supp5 libraries were first designed in J.S. Weissman's laboratory by splitting their original 10 sgRNAs/gene library based on predicted sgRNA knockdown activity (Horlbeck et al., 2016). As expected, the Top5 sgRNAs yielded more hits than Supp5 sgRNAs (Fig. S3 d). In addition, the Top5 sgRNA library behaves similarly to the original 10 sgRNAs/gene library, suggesting that five efficient sgRNAs are sufficient for hit identification using our imaging-based screening approach. Even in the scenario of Top5 sgRNAs for two runs, ~20 hits were successfully identified (Fig. S3 d). Thus, based on factors such as the time to run a screen and available sgRNAs, fewer sgRNAs/gene and/or fewer runs can be used in a screen.

Because nuclear size often positively correlates with DNA content and cell size, we also sorted cells based on H2B-mGFP intensity (as a proxy for DNA content) or forward scattering (FSC) signal (cell size; Figs. 4 f and S4). To compare results

directly, these two screens were performed at the same time as our imaging-based nuclear size screen (Fig. 4 f). The top 10th percentile of cells based on either GFP fluorescence or FSC signal were separately sorted and prepared for high-throughput sequencing. In the H2B-mGFP intensity screen, two replicates identified 11 and 16 hits, respectively, with 7 in common, while 7 and 0 were identified in the FSC screen (Fig. S4). Together, a total of 21 genes were captured in the H2B-mGFP and FSC screens (Fig. 4 g); 15 of these 21 genes were also identified through the imaging-based nuclear size screen. These data suggest that direct measurement using a microscope can provide different information and reveal hits that are inaccessible using other screening approaches.

## Genes involved in nuclear size regulation

We applied optical enrichment to a screen for genes involved in nuclear size determination and identified 15 hits. To validate the 15 genes that emerged in both replicates of the microscope-based screen for enlarged nuclei, each gene was individually targeted using the sgRNAs from our sgRNA library. 11 of 15 genes showed >75% knockdown, as revealed by real-time quantitative PCR (RT-qPCR), with most genes demonstrating almost complete knockdown (Fig. S5). Furthermore, 14 of 15 hits were confirmed to be real hits (Kolmogorov–Smirnov test two-tailed P < 0.01 for at least two of three replicates; the exception was TACC3, which could be explained by incomplete knockdown; Figs. 5 a, S5, and S6). Among the 14 verified genes, all have known roles during cell cycle regulation except KRI1, which is involved in cell death regulation in Caenorhabditis elegans (Ito et al., 2010; Data S3). Six genes are involved in spindle function and chromosome segregation, which includes KIF11 (Rapley et al., 2008), NUP62 (Hashizume et al., 2013), SPDL1 (Gassmann et al., 2008), and three core chromosomal passenger complex components IN-CENP, AURKB, and CDCA8 (Terada, 2001; Carmena et al., 2012). Three genes function in DNA damage and repair, namely TICRR (Sansam et al., 2010; Yu et al., 2019), TOP2A (Bower et al., 2010; Yoshida and Azuma, 2016), and RAD51 (Yoon et al., 2014; Sullivan and Bernstein, 2018), while the remaining four play roles in histone synthesis (CASP8AP2; Sokolova et al., 2017), DNA maintenance (DNA2; Duxin et al., 2009; Pawłowska et al., 2017), and cell cycle regulation (SKA1; Sivakumar et al., 2014, 2016; and FBXO5; Verschuren et al., 2007; Machida and Dutta, 2007; Data S3). Some of these functions might directly explain the larger nuclei phenotype after knockdown. For example, the loss of EMI1 protein (product from FBXO5) was suggested to lead to cellular senescence, resulting in larger nuclei (Verschuren et al., 2007). Knockdown of chromosomal passenger complex components (product from AURKB, INCENP, and CDCA8) leads to asymmetrical distribution of nuclear material and cytokinesis failure, thereby generating abnormally large nuclei (Terada, 2001; Carmena et al., 2012).

To begin to understand the mechanism underlying nuclear size regulation of our 14 hits, we investigated changes in DNA content, measured by DRAQ5 staining, and cell size, assessed using FSC on FACS, after knockdown. Almost all hit genes show increases in the FSC signal (Figs. 5 b and S6). This matches with the karyoplasmic ratio theory, which suggests that nuclear size

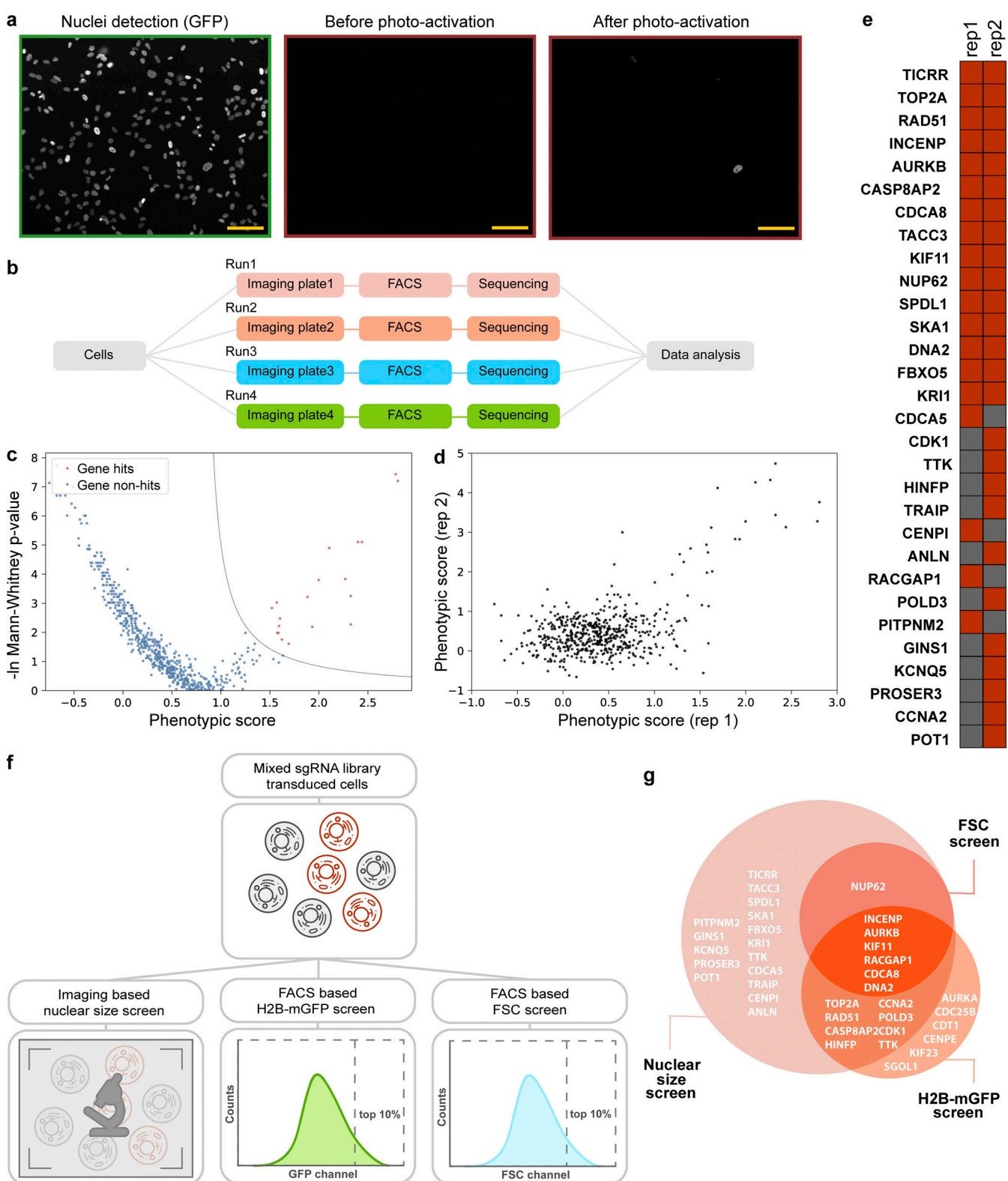

Figure 4. **Screens for nuclear size regulators. (a)** Example images of the nuclear size screen. Cells (hTERT-RPE1 dCas9-KRAB-BFP PA-mCherry H2B-mGFP) were transduced with a CRISPRi sgRNA library and imaged under the GFP channel. Cells with nuclei >1,000 µm² were photoactivated, sorted, and analyzed by deep sequencing. Example images of nuclei, (GFP channel, green), before and after photoactivation (mCherry channel, red). Note that the example images were from experiments done with dual-activation setup as described in Fig. 3 c. Background cells with low fluorescence intensity in mCherry channel after photoactivation were true negative cells that were photoactivated with a shorter exposure time (200 ms). Scale bar: 100 µm. **(b)** Workflow of one replicate of the nuclear size screen. For each replicate, transduced cells were seeded into four glass-bottom imaging plates. Cells in each single imaging plate were imaged, analyzed, photoactivated, sorted, and sequenced separately, termed as separate runs. **(c)** Screening result of one replicate shown in volcano plot. The

corresponding eFDR-η curve is shown in Fig. S3 b, and the other replicate is shown in Fig. S3 c. **(d)** Comparison between two replicates. **(e)** List of genes identified from two replicates. Red, hit; gray, non-hit. **(f)** Workflow of three screens, namely nuclear size screen, H2B-mGFP screen, and FSC screen. After transducing the sgRNA library, cells were split and prepared for either image analysis (nuclear size screen) or FACS sorting (H2B-mGFP screen and FSC screen). **(g)** Comparison of hits identified in FSC screen, H2B-mGFP screen, and nuclear size screen.

is always related to cellular size (Webster et al., 2009; Edens et al., 2013; Mukherjee et al., 2016; Cantwell and Nurse, 2019). On the other hand, DRAQ5 signal was unchanged or somewhat lower after knockdown (Figs. 5 b and S6), suggesting that these gene knockdowns do not change cellular DNA content.

## Discussion

High-throughput sequencing has transformed our ability to perform pooled genetic screens on a broad scale. However, applying enrichment-based pooled CRISPR screens to optical-based phenotypes has been challenging. In this study, we developed an imaging-based pooled CRISPR screening method. Using the photoactivatable fluorescent protein PA-mCherry, cells of interest can be labeled through photoactivation and isolated with FACS sorting, which enables sgRNA identification by high-throughput sequencing. We have combined this optical enrichment strategy with pooled CRISPR-Cas9 libraries to perform imaging-based CRISPR screens. Independently, Kanfer et al. (2020 Preprint) described a similar method to ours for imaging-based pooled CRISPR screening.

### Advantages and limitations of phenotypic screening by optical enrichment

Image processing and microscope operations are the time-limiting steps of most imaging-based genetic screens. Our optical enrichment–based pooled screening method is relatively fast and scalable. For example, the image analysis code developed for this study can be run on a millisecond time scale per field of view, and cells can be separated from the Ctrl population on a FACS machine with as little as 100-ms photoactivation time (Fig. 2 b), enabling screening of large numbers of cells. In our system, 1.5 million RPE-1 cells can be imaged and photoactivated in 8 h with a 20× objective. This is significantly faster than in situ methods, which process millions of cells over a period of a few days (Feldman et al., 2019). For phenotypes as penetrant as mIFP expression, a library of 6,092 sgRNAs with 2 sgRNAs/ group can be successfully screened with a single replicate. A genome-scale screen of such a phenotype can be executed within 3 d (time of image analysis and photoactivation). Even for more complex phenotypes, such as the nuclear size screen described here, a genomic screen can be finished within 2 wk using the Top5 sgRNA library and two runs. This time can be shortened with further optimization such as the use of a microscope with a larger field of view, a lower-magnification objective, optimization of imaging analysis algorithms, etc.

Optical enrichment screening also is possible for phenotypic screens with relatively low hit rates (defined as the fraction of all genes screened that are true hits). The ability to detect hits at low hit rates in our method depends on multiple factors, including (a) the penetrance of the phenotype; (b) cellular fitness

effect of the phenotype; (c) detection and photoactivation accuracy of the phenotype; and (d) limitations imposed by FACS recovery and sequencing sample preparations of low cell numbers. The first three factors vary with the phenotype of interest. We optimized the genomic DNA preparation protocol (Materials and methods) and are now able to process sequencing samples from a few thousand cells, enabling screens of low-hit-rate phenotypes. In our nuclear size screen, >1.5 million cells were analyzed during each run, with 2,000–4,000 cells recovered after FACS sorting. The hit rate of this screen was 2.76%, similar to optical CRISPR screens performed in an arrayed format (de Groot et al., 2018), demonstrating the possibility to apply our approach to investigate phenotypes with low hit rates.

Our optical enrichment screening approach can screen for multiple phenotypes simultaneously by using different photoactivation times. With PA-mCherry, we show that four distinct phenotypes could be potentially sorted (Fig. 2 b). We demonstrate this in practice by differential photoactivation of true-positive and -negative cells to improve screening sensitivity. However, differential time of photoactivation could also be applied to analyze different phenotypes. This approach can be further developed by combining multiple photoactivatable fluorescent proteins in each cell.

In our study, optical enrichment was used for pooled CRISPR screens on phenotypes identifiable through microscopy. However, optical enrichment can be used for other purposes, as demonstrated previously (Hasle et al., 2020). In a recent study by Hasle et al. (2020), the process of separating cells by FACS after optical enrichment was termed "visual cell sorting." This method was used to evaluate hundreds of nuclear localization sequence variants in a pooled format and to identify transcriptional regulatory pathways associated with paclitaxel resistance using single-cell sequencing, demonstrating the broad applicability and power of this approach beyond CRISPR screening.

Our approach has limitations. Phenotypes of interest should be identifiable under the microscope and generally require fluorescent labeling. Commonly used fluorescence microscopes use four channels for fluorescent imaging, with little spectral overlap: blue, green, red, and far red. In our study, the red channel was occupied by cell labeling with PA-mCherry, and the blue channel was used to estimate sgRNA transduction efficiency. Because sgRNA transduction efficiency can be measured by other approaches, the blue channel could be used together with the remaining two channels to label cellular structures. A combination of bright-field imaging with deep learning can be used to reconstruct the localization of fluorescent labels (Ounkomol et al., 2018), making it possible to use bright-field imaging to further expand the phenotypes that can be studied with our technique.

Another limitation is the computational cost. Phenotypes were identified directly after imaging; thus the analysis code has

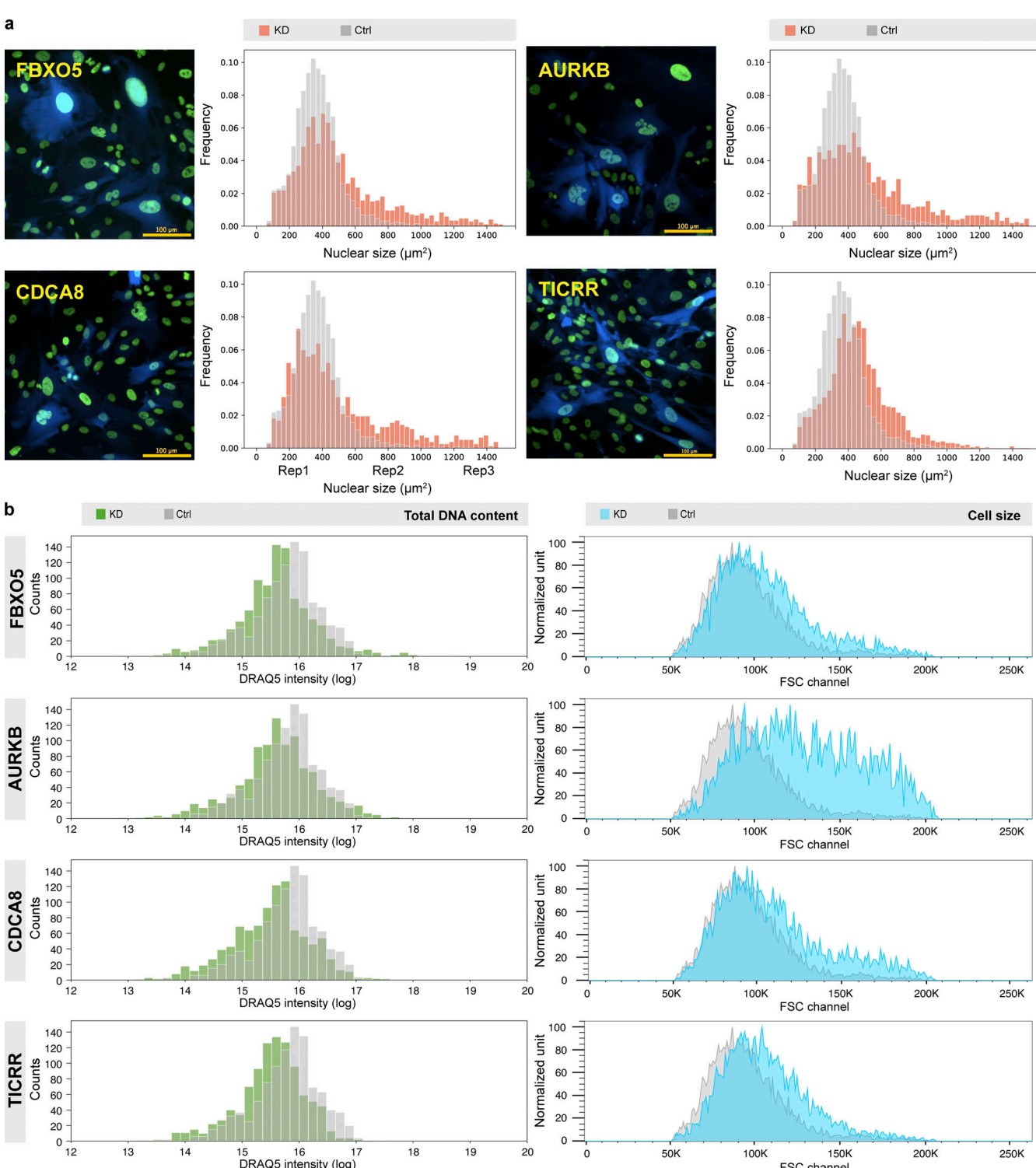

Figure 5. **Characterization of hits identified in nuclear size screen. (a)** Each hit identified in both replicates was verified under the microscope after infecting with a mixture of three to four sgRNA constructs targeting the gene (*n* = 3). Cells were puromycin selected for 2 d before imaging. Example images of four hits and their distribution of nuclear sizes from one replicate are shown in panel a; all the others are listed in Fig. S6. For each gene in each replicate, at least 1,000 cells were analyzed using the Auto-PhotoConverter µManager plugin. The cell population is heterogeneous due to inefficient knockdown, incomplete puromycin selection, and penetrance of the phenotype. A BFP was expressed from the same sgRNA construct. Only cells with high BFP intensity, indicating successful sgRNA transduction, were included for data analysis as described in Materials and methods. Red, nuclear size distribution of corresponding gene after knockdown; gray, nuclear size distribution of cells infected with nontargeting control sgRNAs. **(b)** Some cells developed a larger cellular size while maintaining a similar DNA content level after knockdown. For DNA content measurement, cells were infected with the same three to four sgRNAs/gene, puromycin selected for 2 d, and stained with 5 µM DRAQ5 for 1 h before imaging (1,000 cells were analyzed for each gene). For cellular size measurement, cells were infected with the same three to four sgRNAs/gene and puromycin selected for 3 d before FACS analysis (at least 2,787 cells were analyzed

for each gene). Example imaging analysis data and FACS data of the same four hits are shown in b, and all the others are shown in Fig. S6. Green, distribution analysis of DRAQ5 staining intensity after knockdown of corresponding gene; blue, FACS of FSC signal after knockdown of corresponding gene; gray, distribution analysis of DRAQ5 staining intensity or FSC signal of cells infected with nontargeting control sgRNAs.

to be fast and robust. In our study, the code identified phenotypes within a few hundred milliseconds. Each phenotype required writing specific image analysis code. This requirement can be overcome by implementing other image analysis strategies, including trainable machine learning or combining with existing image analysis software such as CellProfiler, etc., which will benefit laboratories that do not have the expertise to develop custom image-processing code. Additionally, our approach is currently not compatible with fixed cells; thus transient phenotypes might be difficult to capture. However, we expect this to be solvable by further optimizing our screening pipeline to make it possible to prepare sequencing samples after fixation.

### Optical enrichment compared with other methods for phenotypic screening

Two other methods have recently been developed to use imaging for both phenotypic screening and decoding to permit sgRNA identification in individual cells in situ (Feldman et al., 2019; Wang et al., 2019). In both methods, CRISPR sgRNA expression constructs were modified to express both a sgRNA and a barcode. The barcode can be read out by either in situ sequencing (Feldman et al., 2019) or sequential FISH (Wang et al., 2019). Both methods require sgRNAs to be rebarcoded, necessitating de novo design and library resynthesis and preventing reuse of most existing sgRNA libraries. In addition, cells need to be fixed, preventing further cell-based assays of the identified cells. Most importantly, neither of these methods can easily scale to the whole genome because of barcoding limitations and the long imaging time required.

Another newly published method, similar to ours, also uses high-throughput sequencing as an end point assay. Instead of using FACS to enrich cells of interest, this method cultures cells on microcraft arrays (magnetic polystyrene particles designed to capture single clones) to enable cell isolation as separate clones (CRaft-ID; Wheeler et al., 2020). This method also can use most available sgRNA libraries and is compatible with further live-cell studies. However, it is difficult to perform a genome-wide screen with CRaft-ID, since it requires single-cell isolation during cell culture and thus limits the number of cells that can be screened (6,000 colonies/array). In addition, CRaft-ID cannot be used to screen for phenotypes that cause defects in monoclonal growth, including essential genes. Our assay, on the other hand, provides an option for genome-wide screens and allows study of genes essential to growth.

### Conclusion

In summary, our data demonstrate the power of our optical enrichment–based, pooled CRISPR screening method to study previously inaccessible phenotypes with high efficiency and accuracy. This method is simple and fast, uses open-source software, and can be applied to commercial or institutional genome-scale CRISPR sgRNA libraries. A DMD is required, but

this can be introduced into the light path of common commercial microscopes. This screening approach could be broadly applied across many biological phenotypes, including morphological changes, subcellular organization, and cellular dynamics. Pluggable image analysis code enables selection of any desired morphological phenotypes as long as fast and robust detection code can be created, which is an area suited for deep learning approaches. We anticipate that this screening approach can be integrated with other profiling technologies such as single-cell sequencing, further expanding its application to other research fields.

## Materials and methods

### Plasmid sequences

CRISPRi construct (85969; Addgene) and sgRNA parental construct (84832; Addgene) were a kind gift from J.S. Weissman's laboratory. Other plasmid constructs used in this study are described in Data S4.

### Cell line generation

#### hTERT-RPE1 dCas9-KRAB-BFP

All the hTERT-RPE1 cells were grown in DMEM/F-12 supplemented with 10% FBS and penicillin/streptomycin (complete DMEM/F-12). CRISPRi modality dCas9-KRAB-BFP construct was stably expressed in hTERT-RPE1 cells via lentiviral infection, as described below. BFP-positive cells were sorted after 2 d.

#### hTERT-RPE1 dCas9-KRAB-BFP PA-mCherry

The photoactivatable cell line was generated starting with hTERT-RPE1 dCas9-KRAB-BFP cell line. The PA-mCherry construct was stably expressed in hTERT-RPE1 dCas9-KRAB-BFP cells via lentiviral infection as described below. Monoclonal cell lines were grown and screened under the microscope to select clones with successfully integrated PA-mCherry construct. A cell line that showed high and homogeneous fluorescence after photoactivation was chosen to use in this study.

#### hTERT-RPE1 dCas9-KRAB-BFP PA-mCherry H2B-mGFP and hTERT-RPE1 dCas9-KRAB-BFP PA-mCherry H2B-mGFP mIFP-NLS

H2B-mGFP and mIFP-NLS constructs were sequentially integrated into hTERT-RPE1 dCas9-KRAB-BFP PA-mCherry cells via lentiviral infection. GFP-positive cells or GFP/mIFP double-positive cells were selected by FACS 2 d after infection.

### sgRNA sequences

The two negative Ctrl sgRNAs were used in this study and their protospacer sequences (the part of the target sequences) are 5′-GCTGCATGGGGCGCGAATCA-3′ and 5′-GTGCACCCGGCTAGG ACCGG-3′. sgRNA libraries used in this study were gifts from J.S. Weissman's laboratory. Because the cell line used (hTERT-RPE1 dCas9-KRAB-BFP PA-mCherry) for the mIFP proof-of-principle

screen has CRISPRi modality, we used two CRISPRa sgRNA libraries for this screen. These libraries are described in Table S1 and Table S2. The CRISPRi sgRNA library used in the nuclear size screen is described in Table S3. sgRNAs used for hit verification are listed in Table S4.

## Lentivirus preparation and transduction
For CRISPRi modality construct and sgRNA libraries, lentiviral particles were packaged by transfecting HEK293T in a 15-cm cell culture dish at 70% confluence with 8 µg plasmid, 1 µg PMD2.G, 8 µg dR8.91, 48 µl TransIT-LT1 transfection reagent (Mirus Bio), and 1,300 µl serum-free Opti-MEM. Medium containing lentivirus was collected 72 h after transfection and concentrated 10-fold using an Amicon Ultra Centrifugal Unit (MilliporeSigma). For other constructs including PA-mCherry, H2B-mGFP, mIFP-NLS, and small-scale sgRNA virus preparations, lentiviral particles were packaged by transfecting HEK293T in a six-well plate at 70% confluence with 1 µg PA-mCherry plasmid, 0.1 µg PMD2.G, 0.9 µg psPAX2, 10 µl TransIT-LT1 transfection reagent, and 250 µl serum-free Opti-MEM. Medium containing lentivirus was collected 72 h after transfection, and concentration was not needed. 250 µl supernatant was used to transduce a six-well plate of corresponding cells by spinning infection at 2,000 rpm for 1 h. Polybrene infection reagent (Sigma-Aldrich) was used to increase infection efficiency. Medium was replaced with complete DMEM/F-12 immediately after spinning infection. Cells were puromycin selected at 5 µg/ml to select for cells successfully receiving the sgRNA (sgRNA construct harbors puromycin resistance cassette). For screening, cells were puromycin selected for 3 d.

## Microscopy
Cells were grown in 96-well glass bottom dishes (Matriplate; Brooks) after puromycin selection. Images were acquired by fluorescence imaging using a Nikon Eclipse Ti-E microscope with a Nikon 20× 0.75-NA (Plan APO VC) objective. A DMD (DLP LightCrafter 6500 Evaluation Model; Texas Instruments) was positioned behind the back port of the microscope and illuminated using a Sutter HPX-L5UVLambda LED light source (8 W output centered around 405 nm) coupled through a 5-mm-diameter liquid light guide. The DMD image was projected into the sample plane using a 100-mm-focal-length achromatic doublet lens and a 1× beam "expander" consisting of a pair of 80-mm-focal-length achromatic lenses, followed by a 450-nm longpass dichroic mirror positioned on top of the dichroic mirror used for epi-illumination (Fig. S1 a). With all pixels of the DMD in the "on" position, we measured ~40 mW in the back focal plane of the objective. During image acquisition, cells were maintained in DMEM/F-12 complete medium at a constant temperature of 36–37°C using a stagetop incubator (Tokai Hit). Fluorescence illumination was with a liquid light guide coupled LED illuminator (SpectraX; Lumencor) using a multibandpass dichroic mirror (FF410/504/582/669-Di01-25 × 36; Semrock) in a cube with the Semrock FF01-440/521/607/700-25 as emission filter. Camera (Andor Zyla) exposure times were usually set to 500 ms for GFP channel, 100 ms for mCherry channel, and 1,000 ms for mIFP channel.

## FACS
For all the screens requiring cell sorting, cells were trypsinized and sorted using a BD FACSAria III. For hit analysis, cells were analyzed with a BD FACSAria II after 3-d puromycin selection. Cells were gated for single-cell population, and FSC levels were analyzed using FlowJo v10.6.2. Detailed gating strategy is provided in Data S1.

## Sequencing sample preparation
Sequencing sample was prepared using a protocol from J.S. Weissman's laboratory (https://weissmanlab.ucsf.edu/CRISPR/IlluminaSequencingSamplePrep.pdf) except that genomic DNA of samples <10,000 cells was extracted with the Arcturus Pico-Pure DNA Extraction Kit.

## mIFP proof-of-principle screen, nuclear size screen, FSC screen, and H2B-mGFP screen
For the mIFP proof-of-principle screen, mIFP-positive cells (hTERT-RPE1 dCas9-KRAB-BFP PA-mCherry H2B-mGFP mIFP-NLS) and mIFP-negative cells (hTERT-RPE1 dCas9-KRAB-BFP PA-mCherry H2B-mGFP) were stably transduced with the mIFP sgRNA library (CRISPRa library with 860 elements, see Table S1) and the Ctrl sgRNA library (CRISPRa library with 6,100 elements, see Table S2) separately. For the nuclear size screen, FSC screen, and H2B-mGFP screen, cells (hTERT-RPE1 dCas9-KRAB-BFP PA-mCherry H2B-mGFP) were stably transduced with the nuclear size library (CRISPRi library with 6,092 elements, see Table S3). To guarantee that cells received no more than one sgRNA per cell, BFP was expressed on the same sgRNA construct and cells were analyzed by FACS the day after transduction. The experiment continued only when 10–15% of the cells were BFP positive. These cells were further enriched by puromycin selection (a puromycin resistance gene was expressed from the sgRNA construct) for 3 d to prepare for imaging. For FSC and H2B-mGFP screens, cells were then subjected to FACS sorting. Cells before FACS (unsorted sample for FSC and H2B-mGFP screens) and top 10% cells based on either FSC signal (high FSC sample) or GFP fluorescence signal (high GFP sample) were separately collected and prepared for high-throughput sequencing. For mIFP proof-of-principle screen and nuclear size screen, cells were then seeded into 96-well glass-bottom imaging dishes (Matriplate; Brooks) and imaged starting from the morning of the next day (~15 h after plating). A series of densities ranging from $0.5 \times 10^4$ cells/well to $2.5 \times 10^4$ cells/well with an interval of $0.5 \times 10^4$ cells/well were selected and seeded. The imaging dish with cells at ~70% confluence was selected to be screened on the imaging day. For mIFP proof-of-principle screen, a single imaging plate was performed for each replicate, while four imaging plates per replicate were imaged for the nuclear size screen. When executing multiple imaging runs, two consecutive runs could be imaged on the same day (day run and night run). 64 (8 × 8, day run) or 81 (9 × 9, night run) fields of view were selected for each imaging well, and each field of view was subjected to an individual round of imaging directly followed by photoactivation. Approximately 200–250 cells were present in each given field of view, and 60% to 80% surface area of each well was covered. Either mIFP-positive cells or cells

passing the nuclear size filter were identified and photoactivated automatically using the Auto-PhotoConverter μManager plugin. The total time to perform imaging and photoactivation of a single 96-well imaging dish with ~1.5 million cells was ~8 h. The night run generally took longer, since more fields of view were included than in the day run. Cells were then harvested by trypsinization and pooled into a single tube for isolation by FACS. Sorting gates were predefined using samples with different photoactivation times (e.g. 0 s, 200 ms, and 2 s), and detailed gating strategies are described in Data S1. Sorted samples were used to prepare sequencing samples.

### Bioinformatic analysis of the screen
Analysis was based on the ScreenProcessing pipeline developed in J.S. Weissman's laboratory (https://github.com/mhorlbeck/ScreenProcessing; Horlbeck et al., 2016). The phenotypic score (ε) of each sgRNA was quantified as previously defined (Kampmann et al., 2013; Data S5). For the mIFP proof-of-principle screen, the phenotypic score of each group was the average score of two sgRNAs assigned to the group and averaged between two replicates unless otherwise described. For the nuclear size screen, FSC screen, and H2B-mGFP screen, genes were scored based on the average phenotypic scores of the sgRNAs targeting them. For the nuclear size screen, phenotypic scores were further averaged between four runs for each replicate. For the nuclear size screen, FSC screen, and H2B-mGFP screen, sgRNAs were first clustered by transcription start site and scored by the Mann–Whitney $U$ test against 22 nontargeting Ctrl sgRNAs included in the library. Because only 22 Ctrl sgRNAs were included, the significance of hits was assessed by comparison with simulated negative Ctrls that were generated by random assignment of all sgRNAs in the library, and phenotypic scores of these simulated negative Ctrls were scored in the same way as phenotypic scores for genes. A score η that includes the phenotypic score and its significance was calculated for each gene and simulated negative Ctrl. The optimal cutoff for score η was determined by calculating an eFDR at multiple values of η as the number of simulated negative Ctrls with score η higher than the cutoff (false positives) divided by the sum of genes and simulated negative Ctrls with score η higher than the cutoff (all positives). The cutoff score η resulting in an eFDR of 0.1% was used to call hits for further analysis (Data S5). An example analysis is described in detail in Data S5, and raw counts and phenotypic scores for all four screens are listed in Data S6 and Data S7.

### Verification of hits from nuclear size screen
For each hit in the nuclear size screen, the two sgRNAs with the highest phenotypic score in the screen and the two sgRNAs with the highest score predicted by the CRISPRi-v2 algorithm (Horlbeck et al., 2016) were selected and pooled to generate a mixed sgRNA pool of three to four sgRNAs (detailed information in Table S4). Cells (hTERT-RPE1 dCas9-KRAB-BFP PA-mCherry H2B-mGFP) were transduced with pooled sgRNAs targeting each gene and puromycin selected for 2 d to prepare for imaging. Cells were then seeded into 96-well glass-bottom imaging dishes. For DRAQ5 staining experiment, cells were further stained with

5 μM DRAQ5 (Cell Signaling) for 1 h before imaging. Images were collected the next day, and nuclear size and DRAQ5 staining intensity was measured using the Auto-PhotoConverter μManager plugin. To focus on cells with successful transduction, BFP was coexpressed on the sgRNA construct, and only cells with BFP intensity above a threshold value were included in nuclear size measurements. This BFP threshold was established by comparing the average BFP intensity of cells with and without sgRNA transduction (Fig. S3 a).

### RNA extraction and RT-qPCR
Total RNA was extracted using Trizol reagent (Invitrogen) according to the manufacturer's instructions. 2 μg of total RNA was treated with Turbo DNase I (Invitrogen), and 1 μg of treated RNA was used for cDNA synthesis using SuperScript III First-Strand Synthesis SuperMix for qRT-PCR (Invitrogen). For RT-qPCR amplification of corresponding hit genes, an initial amplification using corresponding primers (Data S8) was done with a denaturation step at 95°C for 15 min, followed by 40 cycles of denaturation at 95°C for 30 s, primer annealing at 60°C for 30 s, and primer extension at 72°C for 30 s. RT-qPCR was performed using SYBR Green PCR Master Mix (Applied Biosystems) with a Bio-Rad CFX 96 Real Time System. Reactions were run in triplicate, and the housekeeping gene ACTB was used as an internal Ctrl.

### Data and software availability
The raw and processed data for the high-throughput sequencing results have been deposited in NCBI GEO database with accession no. GSE156623. The plugin Auto-PhotoConverter developed for open-source microscope control software μManager (Edelstein et al., 2014) has been deposited on github (https://github.com/nicost/mnfinder).

### Online supplemental material
Fig. S1 shows a diagram of microscope setup, example images of a photoactivation experiment, and the Auto-PhotoConverter μManager plugin. Fig. S2 shows the distribution of phenotypic scores of all sgRNA groups from mIFP proof-of-principle screen analyzed in four different analysis modes. Fig. S3 includes supporting data for nuclear size screen including a control experiment indicating viral infection will not affect nuclear size, screening result of the other replicate and eFDR-η curves for two replicates, and analysis of the minimum requirements for performing such imaging-based CRISPR screens. Fig. S4 includes screening results of FSC and H2B-mGFP screens and their corresponding eFDR-η curves. Fig. S5 includes knockdown efficiency measurements of hits identified in both nuclear size screen replicates. Fig. S6 includes characterization of hits identified in both replicates of the nuclear size screen other than the four shown in Fig. 5. Data S1 describes the FACS gating strategy used in the screens. Data S2 describes detailed image processing methods. Data S3 describes hits identified in both replicates of the nuclear size screen. Data S4 includes plasmid sequences used in this study. Data S5 shows an example of detailed bioinformatic analysis steps for analyzing nuclear size screen data. Data S6 and Data S7 lists raw counts and calculated

phenotypic scores for all the screens described in this study. Data S8 lists primers used for RT-qPCR experiments. Table S1 and Table S2 describe the sgRNA libraries used in the mIFP proof-of-principle screen. Table S3 describes the sgRNA library used in the nuclear size screen. Table S4 describes the sgRNAs used for verifying hits identified from both nuclear size screen replicates.

## Acknowledgments

We thank Luke A. Gilbert, Martin Kampmann, Jess Sheu-Gruttadauria, Taylor Skokan, Kara McKinley, and all the other Vale lab members for helpful discussions.

This work was supported by the Howard Hughes Medical Institute (NIH R35GM118106).

The authors declare no competing financial interests.

Author contributions: M.E. Tanenbaum conceived of the project with input from R.D. Vale and J.S. Weissman; N. Stuurman developed the Auto-PhotoConverter plugin; N. Stuurman developed image analysis code with input from X. Yan; M.A. Horlbeck designed sgRNA libraries with input from M.E. Tanenbaum; X. Yan, C.R. Liem, and M. Jost cloned the sgRNA libraries; X. Yan and S.A. Ribeiro performed the experiments; X. Yan analyzed the data; X. Yan drafted the manuscript; X. Yan, N. Stuurman, and R.D. Vale edited the manuscript with input from M.E. Tanenbaum and S.A. Ribeiro; and all authors read and approved the final article.

Submitted: 29 August 2020

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

# Supplemental material

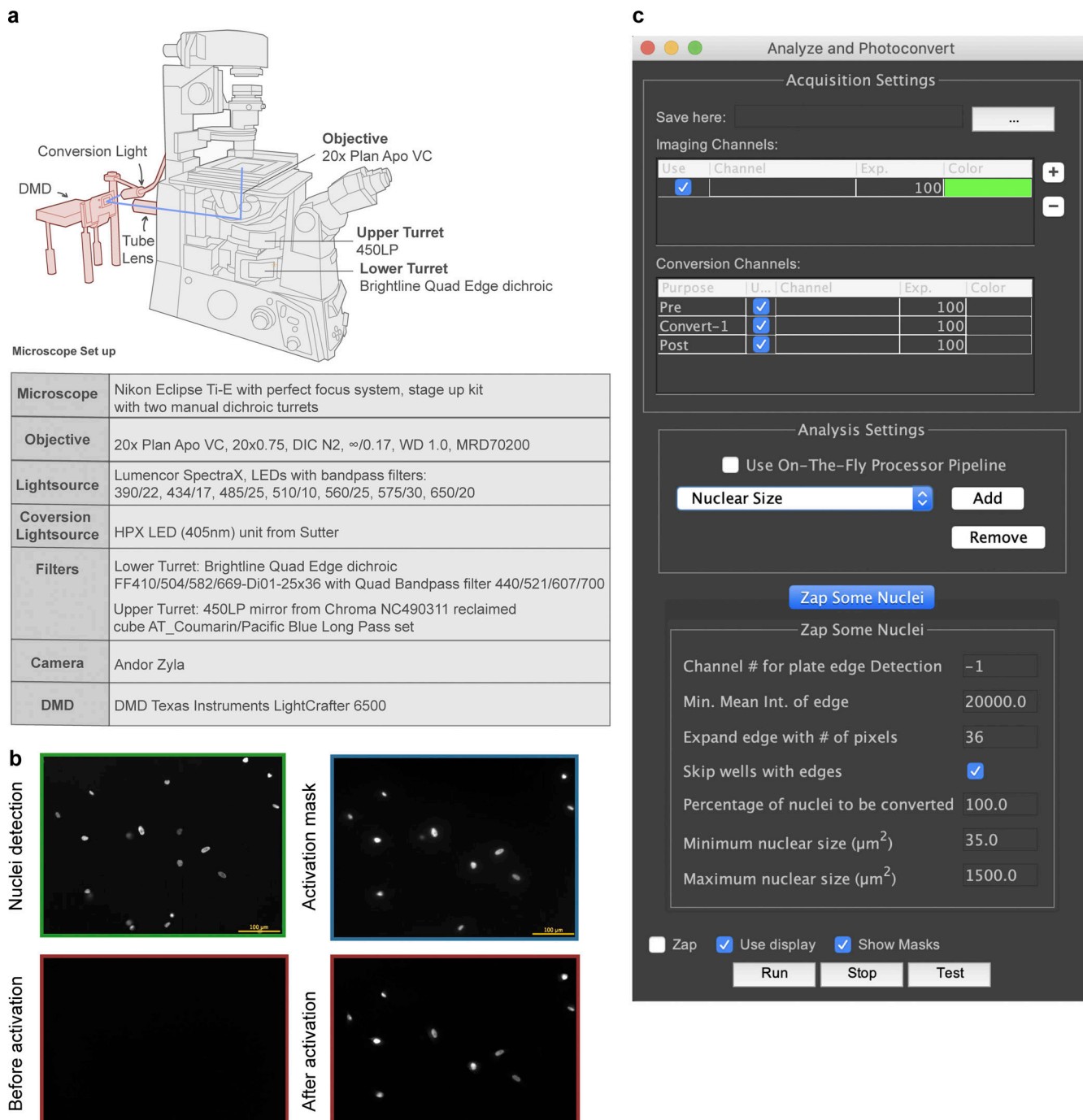

Figure S1. **Microscope and μManager plugin for photoactivation experiments. (a)** A DMD and a blue LED (centered around 405 nm) light source were engineered on a Nikon Eclipse Ti-E microscope as shown in the figure. A computer was used to control the DMD, which reflects light into the microscope only when pixels are in the "on" position, so displaying a mask matching the cell photoactivates that cell. **(b)** Example images of a photoactivation experiment. Cells (hTERT-RPE1 PA-mCherry H2B-mGFP) are shown imaged in the GFP channel (green), during photoactivation (blue light channel, blue), and before and after photoactivation (mCherry channel, red). Scale bar: 100 μm. **(c)** A μManager plugin was developed to enable automatic image acquisition, analysis, and photoactivation. An analysis plugin defines its own set of parameters that can be manipulated by the user. Two analysis plugins were used in this study, one for cell identification and another for nuclear size measurement.

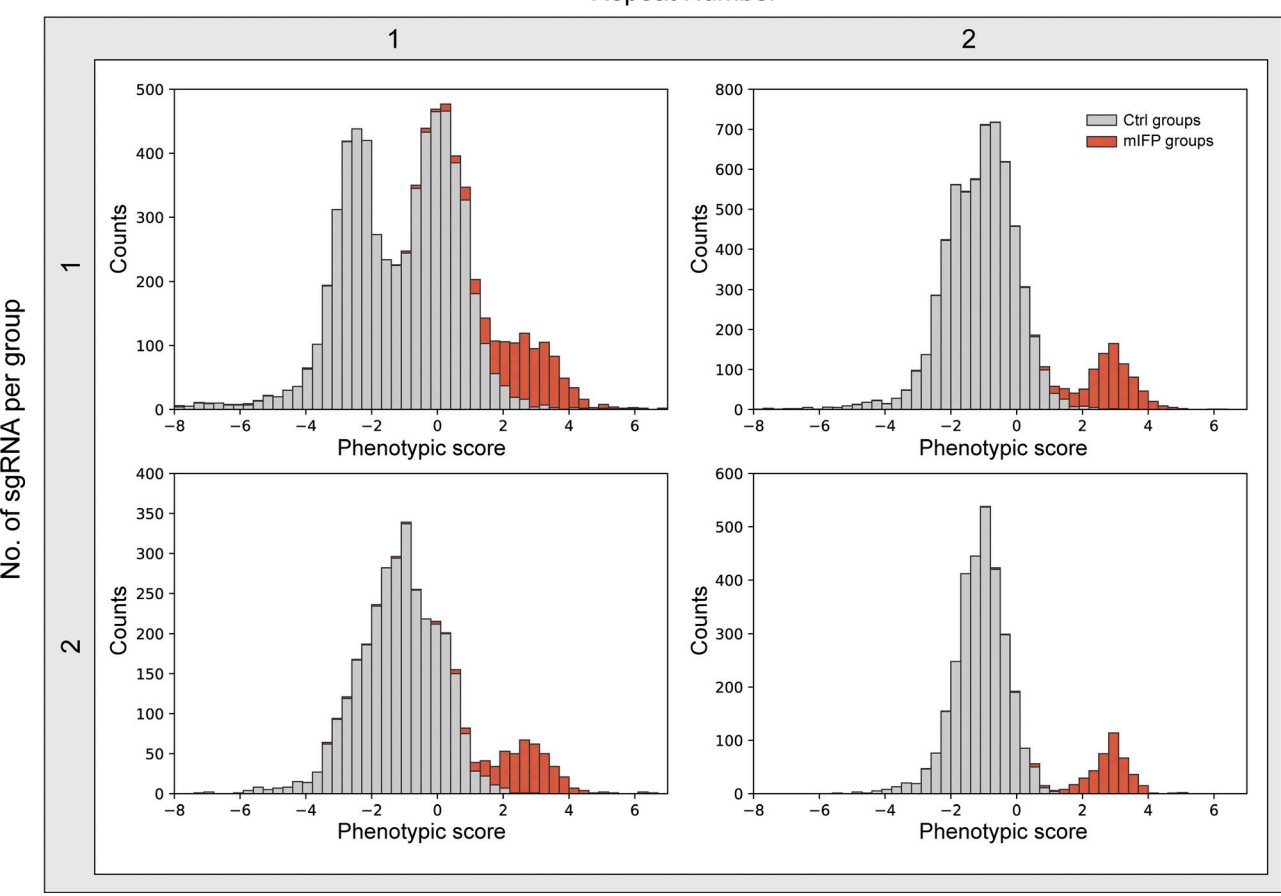

Figure S2. **Library composition and number of replicates affect screening results.** Distribution of phenotypic scores of all sgRNA groups in four different analysis modes. Phenotypic score of a sgRNA group was calculated based on the average phenotypic scores as follows. Top left: A single sgRNA from a single replicate. Top right: A single sgRNA averaged between two replicates. Bottom left: Two sgRNAs from a single replicate. Bottom right: Two sgRNAs averaged between two replicates.

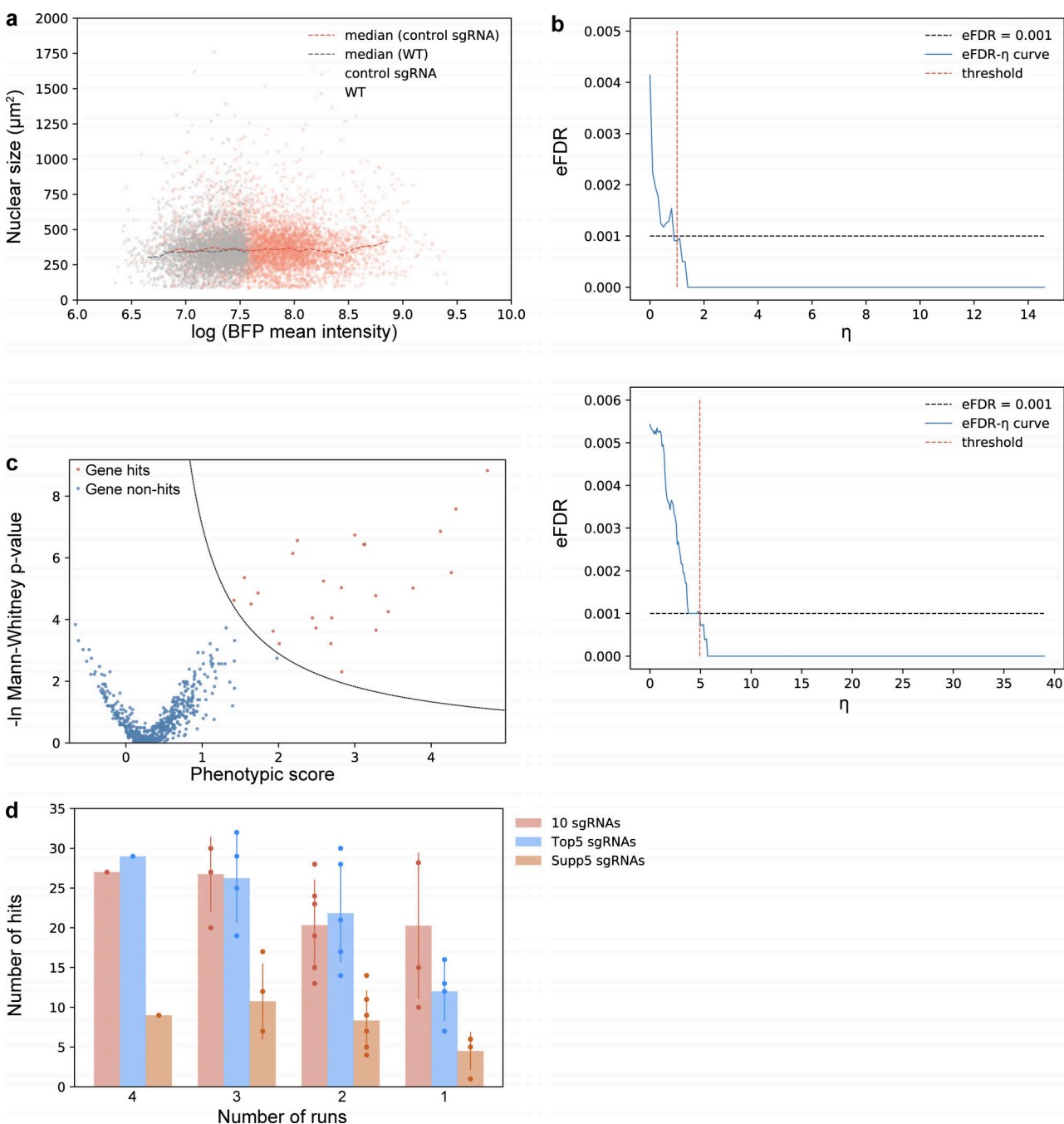

Figure S3. **Screens for nuclear size regulators. (a)** sgRNA transduction results in cells with higher BFP intensity, and negative control sgRNAs do not affect nuclear size after viral infection. Two negative control sgRNAs were designed to have no target sites in the human genome. Cells (hTERT-RPE1 dCas9-KRAB-BFP PA-mCherry H2B-mGFP) underwent viral transduction and puromycin selection for 3 d before imaging. Both wild-type (WT) cells without viral transduction (gray dots), and cells infected with negative control sgRNAs (red dots) were seeded into a 96-well glass-bottom imaging dish. Images were collected for cells with/without sgRNA viral transduction, and both nuclear size and mean BFP intensity of each nucleus were analyzed using the Auto-PhotoConverter μManager plugin (number of wild-type cells analyzed = 2,756; number of negative control sgRNA infected cells analyzed = 5,653). Besides the BFP expressed from the dCas9 construct, another BFP was encoded on the sgRNA construct, and higher BFP intensity was used as a marker for successful infection. The boundary measured from comparison between sgRNA-infected cells and wild-type cells: ln(mean BFP intensity) = 7.6 was also used as a threshold to determine which cells were successfully transduced with sgRNA. Analysis from imaging data shows no correlation between nuclear size and BFP intensity. **(b)** eFDR-η curve for screening result shown in Fig. 4 c. A score η summarizing effects from both severity of the phenotype (phenotypic score) as well as trustworthiness of the phenotype [−ln(P value)] was calculated for each gene and simulated negative control. The optimal cutoff for score η (red dotted line) was determined by calculating an eFDR at multiple values of η as the number of simulated negative controls with score η higher than the cutoff (false positives) divided by the sum of genes and simulated negative controls with score η higher than the cutoff (all positives). The cutoff score η resulting in an eFDR of 0.1% (black dotted line) was used to call hits for further analysis. An example analysis is described in detail in Data S5. **(c)** Screening result of the other replicate shown in volcano plot and its corresponding eFDR-η curve as described in Fig. S3 b. **(d)** Number of hits identified using data averaging using different numbers of runs and/or different library compositions. Error bar: SD.

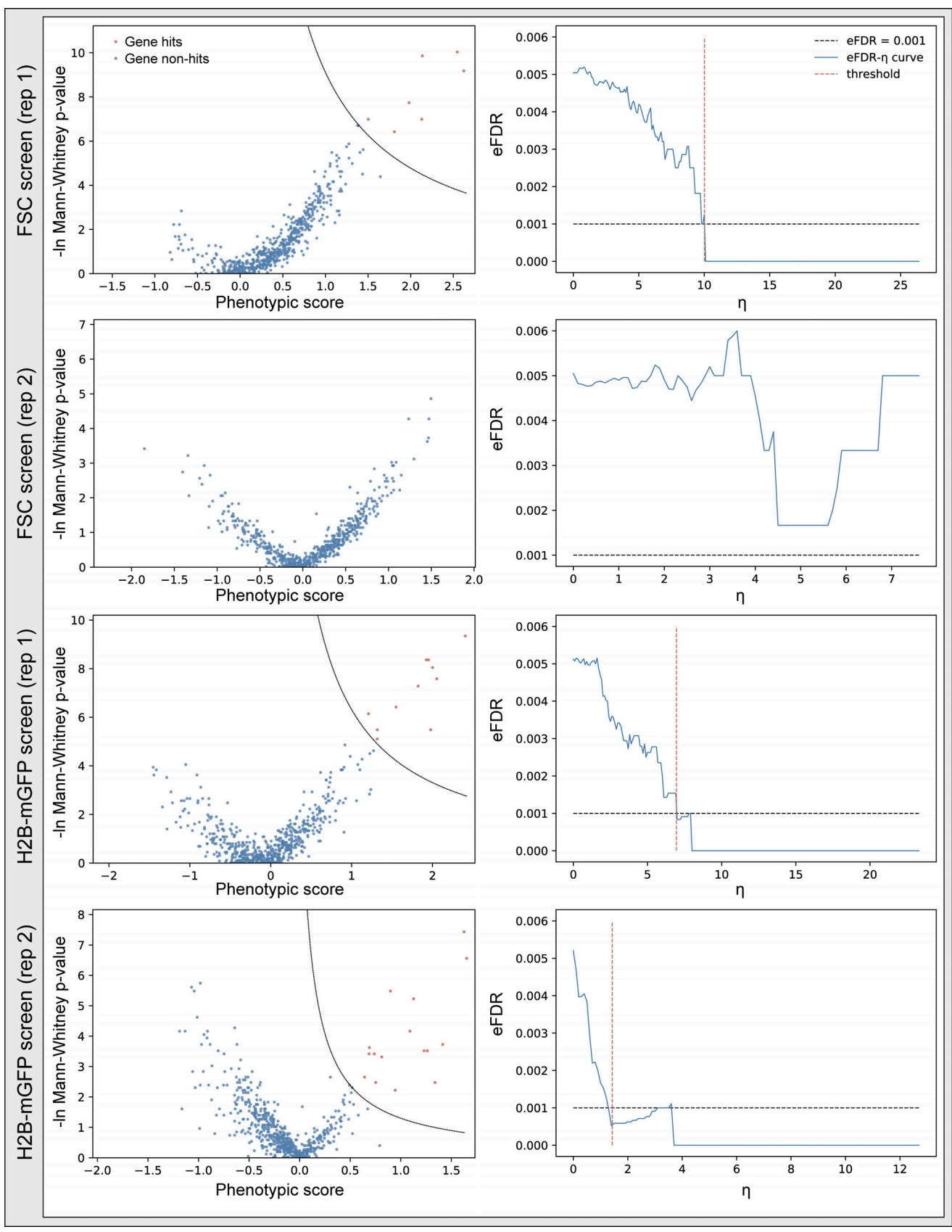

**Figure S4. Screen results of FSC and H2B-mGFP screens.** Cells (hTERT-RPE1 dCas9-KRAB-BFP PA-mCherry H2B-mGFP) were infected and puromycin selected for 3 d. The top 10th percentile of cells based on either GFP fluorescence or FSC signal were separately sorted and prepared for high-throughput sequencing. Screen results shown in volcano plot and their corresponding eFDR-η curve of two replicates as described in Fig. S3 b and Data S5.

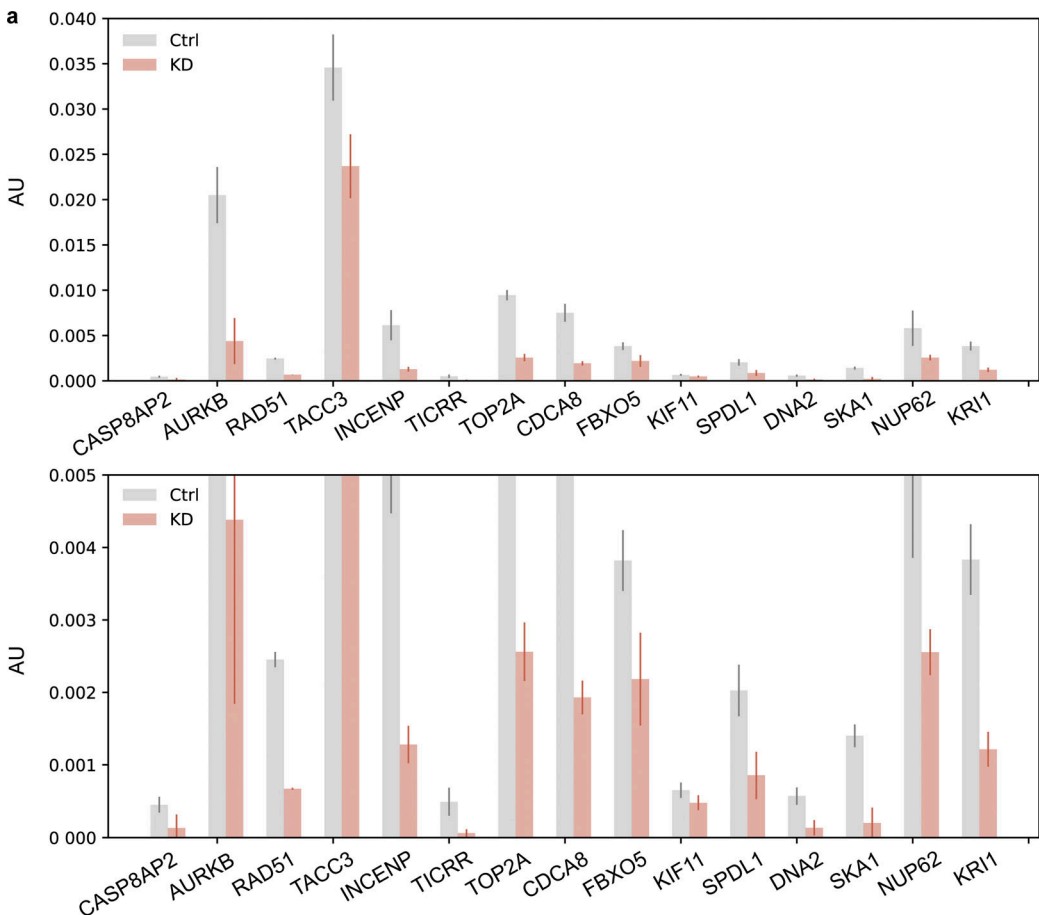

**b**

| Gene | KD percentage | Infection percentage | KD efficiency |
|---|---|---|---|
| **CASP8AP2** | 70.60 | 67.3 | 105 |
| **AURKB** | 78.63 | 51.1 | 154 |
| **RAD51** | 72.69 | 67.9 | 107 |
| **TACC3** | 31.50 | 85.7 | 36.8 |
| **INCENP** | 79.14 | 73.0 | 108 |
| **TICRR** | 87.27 | 80.9 | 108 |
| **TOP2A** | 72.93 | 74.0 | 98.5 |
| **CDCA8** | 74.29 | 66.9 | 111 |
| **FBXO5** | 42.86 | 76.6 | 55.95 |
| **KIF11** | 26.37 | 69.7 | 37.8 |
| **SPDL1** | 57.77 | 77.3 | 74.7 |
| **DNA2** | 76.25 | 79.5 | 95.9 |
| **SKA1** | 85.56 | 80.2 | 107 |
| **NUP62** | 55.97 | 64.5 | 86.8 |
| **KRI1** | 68.29 | 82.4 | 82.9 |

Figure S5.  **Measurement of knockdown efficiency for hit verification. (a)** RT-qPCR results of all the hits identified in both replicates after knockdown. An enlarged graph of the bottom part of the original graph is also included. Cells (hTERT-RPE1 dCas9-KRAB-BFP PA-mCherry H2B-mGFP) were infected with corresponding sgRNAs (Table S4) and puromycin selected for 3 d before harvesting. Harvested cells were split, and one was used for RNA extraction and RT-qPCR analysis to measure the percentage of knockdown (KD percentage), while the other half was used for FACS analysis to measure the percentage of sgRNA infection (Infection percentage; Fig. S5 b). ACTB was used as an internal control to normalize the variability on expression levels. Error bar: SD between triplicates. AU, arbitrary units. **(b)** Knockdown efficiency (KD efficiency) of all the hits identified in both replicates. Knockdown percentage (KD percentage) was measured based on RT-qPCR results. BFP was coexpressed on the sgRNA construct, and only cells with BFP intensity above a threshold value determined by control cells were considered successfully infected cells. Percentage of successful infection (Infection percentage) was measured by FACS and for each gene, and KD efficiency was calculated using KD percentage divided by its corresponding infection percentage.

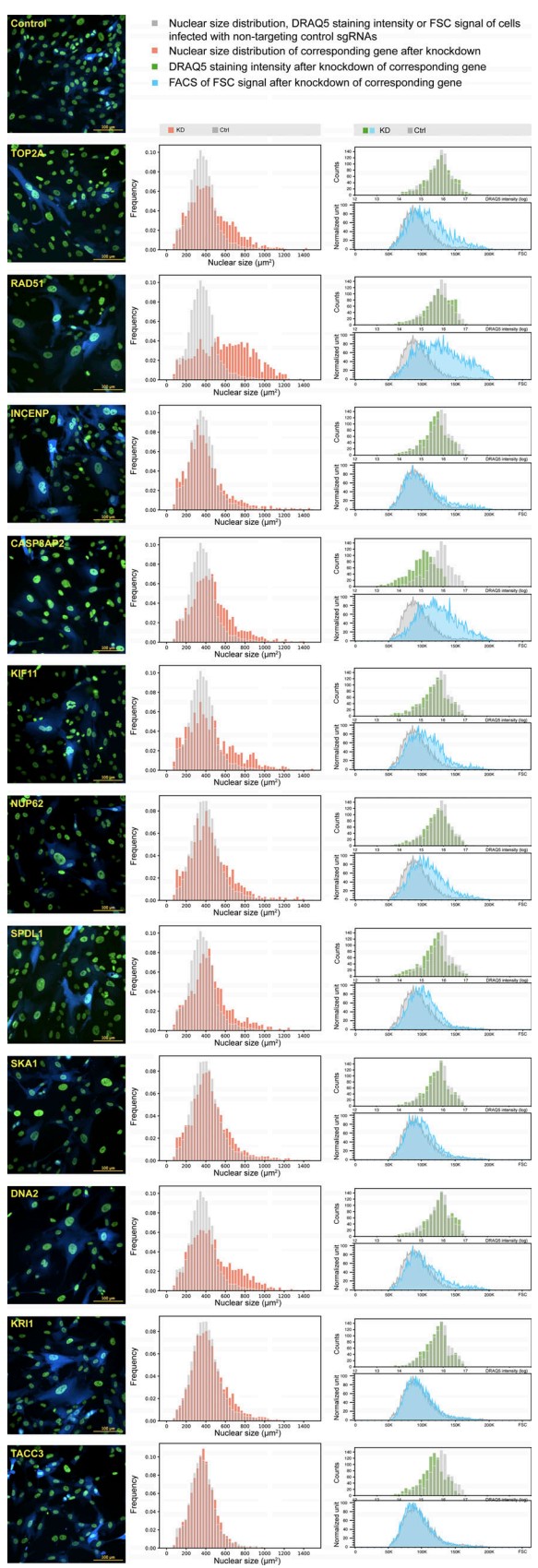

Figure S6. **Characterization of hits identified in both replicates of the nuclear size screens.** Example images, distribution of nuclear size (at least 1,000 cells analyzed for each gene, replicate number = 3), distribution analysis data of DRAQ5 staining fluorescence (1,000 cells analyzed for each gene), and FACS of FSC distribution of each hit (at least 2,787 cells analyzed for each gene; other than the four shown in Fig. 5) after knockdown.

Data S1 describes the FACS gating strategy used in the screens. Data S2 describes detailed image processing methods. Data S3 describes hits identified in both replicates of the nuclear size screen. Data S4 lists plasmid sequences used in this study. Data S5 shows an example of detailed bioinformatic analysis steps for analyzing nuclear size screen data. Data S6 and Data S7 list raw counts and calculated phenotypic scores for all the screens described in this study. Data S8 lists primers used for RT-qPCR experiments. Table S1 and Table S2 describe the sgRNA libraries used in the mIFP proof-of-principle screen. Table S3 describes the sgRNA library used in the nuclear size screen. Table S4 describes the sgRNAs used for verifying hits identified from both nuclear size screen replicates.

