## [Peer Review File · The Journal of Cell Biology]

High-Content Imaging-Based Pooled CRISPR Screens in Mammalian Cells

Xiaowei Yan, Nico Stuurman, Susana Ribeiro, Marvin Tanenbaum, Max Horlbeck, Christina Liem, Marco Jost, Jonathan Weissman, and Ronald Vale

Corresponding Author(s): Ronald Vale, Janelia Research Campus

Review Timeline:	Submission Date:	2020-08-29
	Editorial Decision:	2020-10-21
	Revision Received:	2020-11-17

Monitoring Editor: Jodi Nunnari

Scientific Editor: Tim Spencer

Transaction Report:

DOI: <https://doi.org/10.1083/jcb.202008158>

Revision 0

Review #1

1. How much time do you estimate the authors will need to complete the suggested revisions:

Estimated time to Complete Revisions (Required)

(Decision Recommendation)

Between 1 and 3 months

2. Evidence, reproducibility and clarity:

Evidence, reproducibility and clarity (Required)

In this manuscript Yan et al describe a method to perform imaging based pooled CRISPR screens based on photoactivation followed by selection and sorting of the cells with the desired phenotypes. They establish a system in mammalian RPE-1 cells where they integrate a photo-activatable mCherry, identify the cells of interest under the microscope based on a phenotype, automatically activate the mCherry fluorescence in these cells and then sort the desired populations by FACS. They demonstrate the reliability of their enrichment method and finally use this approach to look for factors that regulate nuclear size by a targeted pooled CRISPR screen. ****Major points:**** 1. This year Hassle et al described a very very similar approach that they name: Visual Cell Sorting . In this case, they use a photoconvertible fluorescent protein (green-to-red conversion) to select cells with a certain visual cellular phenotype and enrich those by FACS. The Hassle et al 2020 MSB paper is only mentioned together with the other methods in the introduction in one sentence (ref #19 in this manuscript): " Recently, several in situ sequencing^{15,16} and cell isolation methods¹⁷⁻²⁰ were developed which allow microscopes to be used for screening. However, these methods contain non-high throughput steps that limit their scalability." I think the current citation of the Hassle et al paper, is not really fair. The idea and the execution of the two approaches are almost exactly the same. Here, the authors concentrate on a CRISPR based application, but obviously the applications of the method are not limited to that. The authors should discuss how these similar ideas can be used in several different applications. 2. While I understand that the authors mean conversion from the dark state to fluorescent state when they describe their photo-activatable mCherry, I think the term "photo-activation" can be confusing for the general reader since typically photo-conversion refers to a change in color. I would here suggest stick to the term photo-activation. 3. For validation of the hits coming from the nuclear size screen: Did the authors have any controls making sure that the right targets were down-regulated? This might be obvious for some of the targets (e.g. CPC proteins that are known to induce division errors display the nuclear fragmentation that the

authors also observe) but especially for the ones that are less known or unknown to induce any nuclear size change, it will be important to demonstrate the specificity of the targets. In addition, it is not clear from the figure legends and the material and methods if these phenotypes are verified by 3-4 gRNAs they use in the validation. Are the histograms representative of a single experiment with one gRNA or a combination of gRNAs in different experiments? Methods of replication of the data presented in Fig4 is unclear. **Minor points:** 1. Related to major point #3: I could not find much experimental info on how the hits from the screen were verified in materials and methods. 2. The legend of Figure 4c is not describing what the plot is showing. Instead it tells the readers the authors' interpretation of the data. 3. Figure S1b there is a typo

3. Significance:

Significance (Required)

I think the idea of performing pooled screens coupled to microscopy is exciting and this approach has definitely more potential than the Craft-ID approach that the authors also discuss in their manuscript. In addition, the approach that is described in this manuscript is convincing and although the fact that the analysis part will require more work (to adapt the software to recognise different types of phenotypic readouts) in the future to make it accessible to the scientific community, the authors present sufficient evidence that the system can be robust. They also present some clever ideas such as to calculate enrichments with different photo-activation times (2sec vs 100ms) followed by separation of these populations by FACS.

Review #2

1. How much time do you estimate the authors will need to complete the suggested revisions:

Estimated time to Complete Revisions (Required)

(Decision Recommendation)

Between 1 and 3 months

2. Evidence, reproducibility and clarity:

Evidence, reproducibility and clarity (Required)

In this manuscript, Yan et al. present optical enrichment, a method for conducting pooled optical screens. Optical enrichment works by combining microscopy to mark cells of interest using the PA-mCherry photo-activatable fluorescent protein with FACS to recover them. The method is similar to other methods (Photostick, Visual Cell Sorting), and provides an alternative to in situ

sequencing/FISH methods. The authors use optical enrichment to conduct a pooled optical CRISPRi screen for nuclear size. They identify and exhaustively validate hits, showing that optical enrichment works for its intended purpose. The development of a uManager protocol and discussion of the number of sgRNA's required for a genetic screen using optical enrichment were welcome. The authors' reported throughput of 1.5 million cells per eight hour experiment is impressive; and the demonstrated use of low cell number input for next generation sequencing appears promising. Overall, the manuscript is well written, the methods clear and the claims supported by the data presented. ****General comments**** -I found the analysis and scoring methods to be lacking, both in terms of the clarity of description and in terms of what was actually done. The authors might consider using established methods (eg <https://www.biorxiv.org/content/10.1101/819649v1.full>). In any case, they should revise the text to clarify what was done and address the other concerns raised below. -Relatedly, details regarding how to perform the experiments described are lacking. It is not clear from the text, figures, "Online Methods" section, and Supplementary Files whether all imaging is performed before activation, or whether each field of view is subject to an individual round of imaging followed by activation. It is also unclear whether cells in 96 well plates are sorted as 96 separate tubes or pooled into a single tube prior to sorting. Furthermore, at a minimum, the following details are requested for each optical enrichment "run". These details are critical considerations for those who seek to use optical enrichment in their own laboratories: • Seeding density • Time elapsed (in hours) between cell plating and optical enrichment • The number of fields of view examined • The median number of cells per field of view; the proportion of each plate's surface area that is imaged and photo-converted • The total time taken (in hours) to perform imaging and photoconversion • The gating protocol used for sorting by FACS (preferably including a figure with example gates for one or two experiments). The gating protocol is described for the genetic screen but not for the control experiments. -The authors use PA-mCherry. There are a variety of other photo-activatable fluorophores available, and it would be good for them to comment on why they chose PA-mCherry. Also, since the method is supposed to be used for generic pooled optical screens, it would be good for the authors to comment on what colors remain available for imaging cellular structures. -In general, the figures are hard to read, with most space being dedicated to beautiful but complex schematics/workflows. Points and fonts should be bigger, and the authors should consider revising the schematics to take up less space. -There is extensive use of editorializing adverbs. Adverbs such as "highly" (abstract and page 15), "easily" (pages 4 and 11), "completely" (page 11), and "only" (page 12) are unnecessary at best and unsupported by the data at worst (e.g. cells are not "completely" separable with 100 ms photo-conversion, see page 11 and Figure 1C). Please remove "completely" from page 11 and consider removing other adverbs as well. -Apologies if I missed it, but I couldn't find a data availability statement. Sequencing reads from the experiments should be deposited in SRA or GEO and made available upon publication. ****Specific comments**** Pages 5/6 - The authors present experiments that show that optical enrichment is highly specific for desired cells. But, they should consider presenting precision (fraction of called positives that are true positive) and recall (fraction of all true positives that are called positive) instead. I think these relate more directly to a pooled optical screen than specificity. Page 6 - Related to the above point, the authors state "These results indicate the assay yields reliable hit identification regardless of the percentage of hits in the library." This statement seems too strong given that the authors looked at specificity experimentally with a mixture of ~1% mIFP positive cells. In fact, hits might be much less than 1% of the total population of cells, and specificity would certainly fall from the 80% measured at

1% of the total population. The authors should do a bit more to fairly discuss their ability to find rare hits. Pages 6/7 - The authors perform a validation experiment using two different sgRNA libraries, infecting mIFP- and mIFP+ cells separately. Then, they demix these populations via optical enrichment, sequence and compute a phenotype score for sgRNAs or groups of sgRNAs. The way the experiment is described and visualized is extremely confusing. If I understood correctly (and I am not sure that I did), the bottom right panel of Figure 2b shows that if sgRNAs are (randomly?) paired AND two replicates are combined then optical enrichment nearly perfectly separates all (combined, paired) sgRNAs in the two libraries. The authors should rewrite this section, especially clarifying what is meant by "1 sgRNA/group and 2 sgRNA/group," and consider changing Figure 2b (perhaps just show the lower right panel?).

Page 8 - Related to Supplementary Figure 3, why are there not clear BFP+ and BFP- populations but instead one continuous population? How was the gating determined (e.g. how was the boundary between red and gray picked)? Here, and generally, flow plots and histograms of flow plots should indicate the number of cells. If replicates were performed, they should be included.

Page 8 - "Nuclear sizes...". The authors should say in the main text what size metric was used.

Page 9 - I am a little confused about the statistical analysis of the screen. In Supplementary File 1, the authors state that p-values were "calculated based on comparison between the distribution of all the phenotypic scores of sgRNAs targeting to the gene/assigning in the group and the one of negative control sgRNAs in the libraries." I presume this means that all phenotypic scores (across replicates) of all sgRNAs targeting each gene were included in a Mann Whitney U test with a single randomized set of phenotypic scores. If that's right, it seems like an odd way to get p-values. Better would be a randomization test, where a null distribution of phenotypic scores for each gene is built by randomizing sgRNA-level scores many times. Then the actual phenotypic score is compared to the randomized null distribution, yielding a p-value. In any case, the authors must clarify what they did in the main text and Supplementary File 1.

Page 9 - It does not appear that the p-values presented in Figure 3c have been adjusted for multiple hypothesis testing. This should be done.

Page 9 - "A value of the top 0.1 percentile of control groups was used as a cutoff for hits." Why? This seems arbitrary. It seems like appropriate false-discovery rate control would enable a more rigorous method for choosing a cutoff.

Page 9 - The same comments regarding analysis and scoring of the optical enrichment screen applies to the FSC and GFP screens.

Page 9 - "These data suggest that a direct measurement utilizing a microscope can provide significant improvement in hit yield even for phenotypes that could be indirectly screened with other approaches." I think this conclusion is too strong. It rests on the assumption that the FSC/GFP phenotypes should have the same set of hits as the microscope phenotype (larger nuclear area). This may not be the case. For example, genes whose inactivation increases GFP expression would be hits in the former, but not latter case. The authors should moderate this statement.

Page 11 - "This is significantly faster than the in situ methods." The authors should provide a citation and an actual comparison to the speed of in situ methods.

Page 12 - I think the authors could say a bit more about the possibility of low hit rate screens. How low do they think it is feasible to go? What hit rates are expected based on existing arrayed optical screens?

Page 14 - It is weird that the discussion includes a fairly important couple of paragraphs that seem to belong in the results (e.g. the text surrounding Figure 4b and c). Obviously, I don't want to prescribe stylistic changes, but I suggest the authors consider moving this description of the experiments/analyses to the results.

Page 14 - The authors validate their hits individually, and observe that expression of hit sgRNAs does increase nuclear size in some cells. But, many/most cells remain control-like in these validation experiments. The authors should comment on why this is the case (e.g.

inefficient knockdown, cell cycle effects, etc). Page 14 - It would be nice to formally compare the control and sgRNA distributions in each panel of 4a and Supplementary Figure 5 (e.g. with a Komolgorov-Smirnov test, etc). That would allow a more precise statement to be substituted for "14 out of 15 hits (the exception was TACC3) were confirmed to be real hits, with cells exhibiting larger nuclei after knock down (Fig. 4a and Fig. S5)," which is not quantitative. Figure 2a - I am not sure it is necessary to show the entire workflow again. The first and possibly last panels are the informative ones here. Figure 3a - Same comment as above - these workflow panels take up a lot of real estate and I suggest simplifying them if possible. Figure 3c - At least on my PDF/screen, the "scrambled control" points appear very light gray and are impossible to find. They should be an easier to spot color. Figure 4b - "Most cells developed a larger cellular size and higher H2B-mGFP level after knock down." I think it would be more accurate to say that the median cell size/GFP level increased, or that some cells developed larger sizes/median GFP levels. Figure 4c - I don't understand "Normalized FITC/nuclear size." Do the bars show the mean/median of a population (if so, why not show a dot plot or box plot or violin plot)? Also, what is FITC (I presume it's GFP levels)? Figure 4c - "Most cells maintained a constant ratio between nuclear size and DNA content..." I'm not sure where DNA content came from. Are the authors assuming that their H2B-mGFP is a proxy for DNA content? Or was some other measurement made? If the former, is there a citable reason why this is a good assumption?

3. Significance:

Significance (Required)

I don't generally comment on significance in reviews. Since ReviewCommons is specifically asking, I'll say that this manuscript describes optical enrichment, a method that is an extension of previous work and is substantially similar to a previously published method, Visual Cell Sorting. However, given the timing, it is obvious that these authors have been working independently on optical enrichment. Since the application is distinct, and optical enrichment incorporates some nice features like software to make it easier to execute, it is clearly of independent value.

Review #3

1. How much time do you estimate the authors will need to complete the suggested revisions:

Estimated time to Complete Revisions (Required)

(Decision Recommendation)

Cannot tell / Not applicable

2. Evidence, reproducibility and clarity:

Evidence, reproducibility and clarity (Required)

This study reports a rapid and high-throughput CRISPR-based phenotypic screen approach consisting of selecting cells with phenotypes of interest, label them by photo-conversion and isolating them by FACS. The idea of the method is interesting (has been around) in principle. The key advantage is that is relatively simple, accessible to many groups as it does not require robotics. However, the manuscript is so badly written and hard to follow, that it makes it difficult to judge the technology, to really understand how the experiments were done and whether the results are interpreted correctly. Strictly speaking, it is unclear whether and how good scientific practices GSP have been followed, as the description of the experiments is sometimes lacking totally. Consequently, it is impossible to seriously evaluate this study and judge whether the technology described is really promising. It is probably less sensitive than arrayed screens, in all likelihood can miss hits that affect growth, cannot capture as many phenotypic classes as one would like from high-content screens and the computational and experimental workflow is more complicated. It is puzzling that the authors don't even compare the results with arrayed screens which are of course the current gold-standard. ****Specific points:**** The specificity test (Fig 1) does not make sense how it is described. If the authors spike a certain percentage of cells that can be photoconverted, when analysing the outcome, there will be three classes: mIFP positive, mIFP/mCherry positive and negative. How can they calculate specificity if they do not know whether they converted all mIFP cells? Also the formula used is questionable or is her an error? Furthermore, it is totally unclear how many cells were used and how they were scanned. If they took 90 negative cells and 10 mIFP cells, getting them all back is easy. If they start with 10×10^9 cells, the specificity should be quantified. Furthermore, the phenotype they pick is an easy and convenient one. Much more challenging is to apply it on a multi-parametric phenotype. Again, this is now the gold standard. In their first sgRNA assay, it is not possible to have a clear idea of what groups they are talking about. Do they mean they get phenotypic signatures which they group? How? They need to describe what they do. Here, only ~3500 genes are scanned (the 6843 is both populations and you only select from the mIFP neg population) and it took them 8hrs. This means for the genome it would require ~60h which is indeed fast. However, this experiment is not clearly described. They cannot select the negative population since there is no fluorescent marker (except false positive which are around 1.7%). So I assume they just randomly pick cells (they should really explain much better what they do!). Why go through the hassle? If these sequences are supposed to be a negative population, just pick them in the computer. Also, they cannot calculate an enrichment compared to the negative population, since two different libraries were infected. Again, I can't follow. I find their results about calculating scores based only on true negatives surprising. The average phenotypic score is improved from 3 to 5, which is enormous. This suggests that the phenotypes induced in the mIFP population are extremely common. These results are hard to interpret given the poor description of the experiment. It is possible that it is the same dataset as in 1, but in that case, the false negatives must be rare since the negatives can be selected by absence of both mCherry and mIFP. In the nuclear size screen, 6000 sgRNAs were screened. To array so many sequences would require 20 plates. They required ~40h for imaging one replicate. This is slow, imagine the time with a 60x lens.

3. Significance:

Significance (Required)

Overall, there is no sufficient evidence in this manuscript to convince this reviewer that this method is valid and truly powerful. I cannot support publication in its present form.

Response to the References

Reviewer #1 (Evidence, reproducibility and clarity (Required)):

In this manuscript Yan et al describe a method to perform imaging based pooled CRISPR screens based on photoactivation followed by selection and sorting of the cells with the desired phenotypes.

They establish a system in mammalian RPE-1 cells where they integrate a photo-activatable mCherry, identify the cells of interest under the microscope based on a phenotype, automatically activate the mCherry fluorescence in these cells and then sort the desired populations by FACS. They demonstrate the reliability of their enrichment method and finally use this approach to look for factors that regulate nuclear size by a targeted pooled CRISPR screen.

****Major points:****

1. This year Hassle et al described a very very similar approach that they name: Visual Cell Sorting . In this case, they use a photoconvertible fluorescent protein (green-to-red conversion) to select cells with a certain visual cellular phenotype and enrich those by FACS. The Hassle et al 2020 MSB paper is only mentioned together with the other methods in the introduction in one sentence (ref #19 in this manuscript):

" Recently, several in situ sequencing^{15,16} and cell isolation methods¹⁷⁻²⁰ were developed which allow microscopes to be used for screening. However, these methods contain non-high throughput steps that limit their scalability."

I think the current citation of the Hassle et al paper, is not really fair. The idea and the execution of the two approaches are almost exactly the same. Here, the authors concentrate on a CRISPR based application, but obviously the applications of the method are not limited to that. The authors should discuss how these similar ideas can be used in several different applications.

We agree with the reviewer that we need to describe more about the Hasle *et al.* paper (now ref #20 in the revised manuscript) and expand our description of other applications that could be performed with the method. For this purpose, we have made the following changes:

We have modified the relevant paragraph in the Introduction.

p.3 the second paragraph

Recently, an imaging based method named "visual cell sorting" was described that uses the photo-convertible fluorescent protein Dendra2 to enrich phenotypes optically, enabling pooled genetic screens and transcription profiling (Hasle, N.; Cooke, A.; Srivatsan, S.; Huang, H.; Stephany, J. J.; Krieger, Z.; Jackson, D.; Tang, W.; Pendyala, S.; Monnat, R. J., Jr.; Trapnell, C.; Hatch, E. M.; Fowler, D. M. 2020). Here, we developed an analogous approach to execute an imaging-based pooled CRISPR screen using optical enrichment by automated photo-activation of the photo-activatable fluorescent protein, PA-mCherry.

We have also added the following paragraph in the Discussion.

p.14 line 1

In our study, optical enrichment was utilized for pooled CRISPR screens on phenotypes identifiable through microscopy. However, optical enrichment can be used for other purposes, as demonstrated previously(Hasle, N.; Cooke, A.; Srivatsan, S.; Huang, H.; Stephany, J. J.; Krieger, Z.; Jackson, D.; Tang, W.; Pendyala, S.; Monnat, R. J., Jr.; Trapnell, C.; Hatch, E. M.; Fowler, D. M. 2020). *In a recent study by Hasle et al.*(Hasle, N.; Cooke, A.; Srivatsan, S.; Huang, H.; Stephany, J. J.; Krieger, Z.; Jackson, D.; Tang, W.; Pendyala, S.; Monnat, R. J., Jr.; Trapnell, C.; Hatch, E. M.; Fowler, D. M. 2020), *the process of separating cells by FACS after optical enrichment was termed “visual cell sorting”. This method was used to evaluate hundreds of nuclear localization sequence variants in a pooled format and to identify transcriptional regulatory pathways associated with paclitaxel resistance using single cell sequencing*(Hasle, N.; Cooke, A.; Srivatsan, S.; Huang, H.; Stephany, J. J.; Krieger, Z.; Jackson, D.; Tang, W.; Pendyala, S.; Monnat, R. J., Jr.; Trapnell, C.; Hatch, E. M.; Fowler, D. M. 2020), *demonstrating the broad applicability and power of this approach beyond CRISPR screening.*

2. While I understand that the authors mean conversion from the dark state to fluorescent state when they describe their photo-activatable mCherry, I think the term "photo-activation" can be confusing for the general reader since typically photo-conversion refers to a change in color. I would here suggest stick to the term photo-activation.

We thank the reviewer for pointing this out and to avoid future confusion, we restricted the usage of photo-conversion to specifically indicate conversion of fluorescence from one color into another: *e.g.* when talking about the published visual cell sorting paper in which Dendra2 is used as a photo-convertible fluorescent protein. We use photo-activation in reference to the activation of PA-mCherry in our work.

3. For validation of the hits coming from the nuclear size screen: Did the authors have any controls making sure that the right targets were down-regulated? This might be obvious for some of the targets (e.g. CPC proteins that are known to induce division errors display the nuclear fragmentation that the authors also observe) but especially for the ones that are less known or unknown to induce any nuclear size change, it will be important to demonstrate the specificity of the targets.

For validating hits coming from the nuclear size screen, we have verified the successful transduction of corresponding sgRNA constructs by FACS analysis, but have not confirmed the knockdown. Before final journal publication, we propose to perform rt-qPCR on our 15 gene hits before and after knockdown to measure the percentage of knockdown separately.

In addition, it is not clear from the figure legends and the material and methods if these phenotypes are verified by 3-4 gRNAs they use in the validation. Are the histograms representative of a single experiment with one gRNA or a combination of gRNAs in different experiments? Methods of replication of the data presented in Fig4 is unclear.

We apologize for the confusion. These phenotypes were verified with pools of 3-4 sgRNAs and the histograms are representative of a single replicate infected with a mixed 3-4 sgRNA pool. We have modified the legend to Figure 5 (original Fig. 4) and the method section to explain this point.

****Minor points:****

1. Related to major point #3: I could not find much experimental info on how the hits from the screen were verified in materials and methods.

The description of the experiment and information about the selected sgRNAs has been added in the Method section as follows:

p. 23

Verification of hits from nuclear size screen

For each hit in the nuclear size screen, the two sgRNAs with the highest phenotypic score in the screen and the two sgRNAs with the highest score predicted by the CRISPRi-v2 algorithm²⁴ were selected and pooled to generate a mixed sgRNA pool of 3-4 sgRNAs (detailed information in Supplementary file 8). Cells (hTERT-RPE1 dCas9-KRAB-BFP PA-mCherry H2B-mGFP) were transduced with pooled sgRNAs targeting each gene and puromycin selected for 2 days to prepare for imaging. Cells were then seeded into 96-well glass bottom imaging dishes. Images were collected the next day and nuclear size was measured using the Auto-PhotoConverter μ Manager plugin. To focus on cells with successful transduction, BFP was co-expressed on the sgRNA construct and only cells with BFP intensity above a threshold value were included in nuclear size measurements. This BFP threshold was established by comparing the average BFP intensity of cells with and without sgRNA transduction (Fig.S3a).

2. The legend of Figure 4c is not describing what the plot is showing. Instead it tells the readers the authors' interpretation of the data.

We agree with this important point and have changed the figure legend of Fig. 5c (original Fig. 4c) to just describe the plot:

***c**, The ratios between median level of nuclear size measured from microscopy and H2B-mGFP fluorescence or FSC signal measured from FACS after knockdown, were plotted separately. TACC3, confirmed to be a control gene, was used for comparison (Grey bar).*

3. Figure S1b there is a typo

The typo has been corrected.

Reviewer #1 (Significance (Required)):

I think the idea of performing pooled screens coupled to microscopy is exciting and this approach has definitely more potential than the Craft-ID approach that the authors also discuss in their manuscript. In addition, the approach that is described in this manuscript is convincing and although the fact that the analysis part will require more work (to adapt the software to recognise different types of phenotypic readouts) in the future to make it accessible to the scientific community, the authors present sufficient evidence that the system can be robust. They also present some clever ideas such as to calculate enrichments with different photo-activation times (2sec vs 100ms) followed by separation of these populations by FACS.

Reviewer #2 (Evidence, reproducibility and clarity (Required)):

In this manuscript, Yan et al. present optical enrichment, a method for conducting pooled optical screens. Optical enrichment works by combining microscopy to mark cells of interest using the PA-mCherry photo-activatable fluorescent protein with FACS to recover them. The method is similar to other methods (Photostick, Visual Cell Sorting), and provides an alternative to in situ sequencing/FISH methods. The authors use optical enrichment to conduct a pooled optical CRISPRi screen for nuclear size. They identify and exhaustively validate hits, showing that optical enrichment works for its intended purpose. The development of a uManager protocol and discussion of the number of sgRNA's required for a genetic screen using optical enrichment were welcome. The authors' reported throughput of 1.5 million cells per eight hour experiment is impressive; and the demonstrated use of low cell number input for next generation sequencing appears promising. Overall, the manuscript is well written, the methods clear and the claims supported by the data presented.

****General comments****

-I found the analysis and scoring methods to be lacking, both in terms of the clarity of description and in terms of what was actually done. The authors might consider using established methods (eg <https://www.biorxiv.org/content/10.1101/819649v1.full>). In any case, they should revise the text to clarify what was done and address the other concerns raised below.

-Relatedly, details regarding how to perform the experiments described are lacking. It is not clear from the text, figures, "Online Methods" section, and Supplementary Files whether all imaging is performed before activation, or whether each field of view is subject to an individual round of imaging followed by activation. It is also unclear whether cells in 96 well plates are sorted as 96 separate tubes or pooled into a single tube prior to sorting. Furthermore, at a minimum, the following details are requested for each optical enrichment "run". These details are critical considerations for those who seek to use optical enrichment in their own laboratories:

- Seeding density
- Time elapsed (in hours) between cell plating and optical enrichment
- The number of fields of view examined

- The median number of cells per field of view; the proportion of each plate's surface area that is imaged and photo-converted
- The total time taken (in hours) to perform imaging and photoconversion
- The gating protocol used for sorting by FACS (preferably including a figure with example gates for one or two experiments). The gating protocol is described for the genetic screen but not for the control experiments.

We agree with the reviewer and apologize for the confusion that arose from our description. We also thank the reviewer for suggesting using established methods. However, MAUDE, an analysis for sorting-based CRISPR screen with multiple expression bins, might not be suitable for our study since 1) the distribution of mCherry fluorescence intensity is a reflection of photo-activation efficiency and not sgRNA effect 2) only one sorting bin is collected for each experimental condition. Our analysis is adapted from an existing method from the Weissman lab (<https://github.com/mhorlbeck/ScreenProcessing>).

We agree with the reviewer regarding clarifying other points and rewrote the following part in the Method section:

p. 20

mIFP proof-of-principle screen, Nuclear size screen, FSC screen and H2B-mGFP screen

For the mIFP proof-of-principle screen, mIFP positive cells (hTERT-RPE1 dCas9-KRAB-BFP PA-mCherry H2B-mGFP mIFP-NLS) and mIFP negative cells (hTERT-RPE1 dCas9-KRAB-BFP PA-mCherry H2B-mGFP) were stably transduced with the “mIFP sgRNA library” (CRISPRa library with 860 elements, see Supplementary file 5) and the “control sgRNA library” (CRISPRa library with 6100 elements, see Supplementary file 6) separately. For the nuclear size screen, FSC screen and H2B-mGFP screen, cells (hTERT-RPE1 dCas9-KRAB-BFP PA-mCherry H2B-mGFP) were stably transduced with the “nuclear size library” (CRISPRi library with 6190 elements, see Supplementary file 7). To guarantee that cells receive no more than one sgRNA per cell, BFP was expressed on the same sgRNA construct and cells were analyzed by FACS the day after transduction. The experiment only continued when 10-15% of the cells were BFP positive. These cells were further enriched by puromycin selection (a puromycin resistance gene was expressed from the sgRNA construct) for 3 days to prepare for imaging. For FSC and H2B-mGFP screens, cells were then subjected to FACS sorting. Cells before FACS (unsorted sample for FSC and H2B-mGFP screens) and top 10% cells based on either FSC signal (high FSC sample) or GFP fluorescence signal (high GFP sample) were separately collected and prepared for high throughput sequencing. For mIFP proof-of-principle screen and nuclear size screen, cells were then seeded into 96-well glass bottom imaging dishes (Matriplate, Brooks) and imaged starting from the morning of the next day (around 15 hr after plating). A series of densities ranging from 0.5E4 cells/well to 2.5E4 cells/well with 0.5E4 cells/well interval were selected and seeded. The imaging dish with cells around 70% confluency was selected to be screened on the imaging day. For mIFP proof-of-principle screen, a single imaging plate was performed for each replicate while 4 imaging plates per replicate were imaged

for the nuclear size screen. When executing multiple imaging runs, 2 consecutive runs could be imaged on the same day (day run and night run). 64 (8x8, day run) or 81 (9x9, night run) fields of view were selected for each imaging well and each field of view was subjected to an individual round of imaging directly followed by photo-activation. Around 200-250 cells were present in each given field of view and 60% to 80% surface area of each well was covered. Either mIFP positive cells or cells passing the nuclear size filter were identified and photo-activated automatically using the Auto-PhotoConverter μ Manager plugin. The total time to perform imaging and photo-activation of a single 96-well imaging dish with around 1.5 million cells was around 8 hr. The night run generally took longer, since more fields of view were included than in the day run. Cells were then harvested by trypsinization and pooled into a single tube for isolation by FACS. Sorting gates were pre-defined using samples with different photo-activation times (e.g. 0s, 200ms, 2s) and detailed gating strategies are described in Supplementary file 1. Sorted samples were used to prepare sequencing samples.

-The authors use PA-mCherry. There are a variety of other photo-activatable fluorophores available, and it would be good for them to comment on why they chose PA-mCherry. Also, since the method is supposed to be used for generic pooled optical screens, it would be good for the authors to comment on what colors remain available for imaging cellular structures.

To address these, we have added the following sentences:

p. 4 line 16

A photo-activatable fluorescent protein was chosen over a photo-convertible fluorescent protein to increase the number of channels available for imaging. PA-mCherry was chosen to leave the better performing green channel open for labeling of other cellular features. Moreover, non-activated PA-mCherry has low background fluorescence in the mCherry channel (Fig. S1b), and it can be activated to different intensities when photo-activated for various amounts of time.

p. 14 line 10

Phenotypes of interest should be identifiable under the microscope and generally require fluorescent labeling. Commonly used fluorescence microscopes use four channels for fluorescent imaging with little spectral overlap: blue, green, red and far red. In our study, the red channel was occupied by cell labeling with PA-mCherry and the blue channel was used to estimate sgRNA transduction efficiency. Since sgRNA transduction efficiency can be measured by other approaches, the blue channel could be used together with the remaining two channels to label cellular structures. Combining bright field imaging with deep learning can be used to reconstruct the localization of fluorescent labels^(Ounkomol, C.; Seshamani, S.; Maleckar, M. M.; Collman, F.; Johnson, G. R. 2018), making it possible to use bright field imaging to further expand the phenotypes that can be studied with our technique.

-In general, the figures are hard to read, with most space being dedicated to beautiful but complex schematics/workflows. Points and fonts should be bigger, and the authors should consider revising the schematics to take up less space.

We thank the reviewer for this remark and revised all figures accordingly. Points and fonts were enlarged, and schematics were simplified or removed.

-There is extensive use of editorializing adverbs. Adverbs such as "highly" (abstract and page 15), "easily" (pages 4 and 11), "completely" (page 11), and "only" (page 12) are unnecessary at best and unsupported by the data at worst (e.g. cells are not "completely" separable with 100 ms photo-conversion, see page 11 and Figure 1C). Please remove "completely" from page 11 and consider removing other adverbs as well.

We agree with the reviewer and the following adverbs have been removed: "highly" in abstract and page 15; "easily" on pages 4 and 11; "completely" on page 11 and three "only" on page 12.

-Apologies if I missed it, but I couldn't find a data availability statement. Sequencing reads from the experiments should be deposited in SRA or GEO and made available upon publication.

We apologize that we missed this, and the sequencing data has been deposited to GEO (GSE156623) which will be made available before final publication. The following part has been added to address this.

p. 24

DATA AND SOFTWARE AVAILABILITY

The raw and processed data for the high throughput sequencing results have been deposited in NCBI GEO database with the accession number (GSE156623). The plugin Auto-PhotoConverter developed for open source microscope control software μ Manager^(Edelstein, A. D.; Tsuchida, M. A.; Amodaj, N.; Pinkard, H.; Vale, R. D.; Stuurman, N. 2014) has been deposited on github (<https://github.com/nicost/mnfinder>).

****Specific comments****

Pages 5/6 - The authors present experiments that show that optical enrichment is highly specific for desired cells. But, they should consider presenting precision (fraction of called positives that are true positive) and recall (fraction of all true positives that are called positive) instead. I think these relate more directly to a pooled optical screen than specificity.

We apologize for our poor terminology. Our original definition of "specificity" is the same as "precision" suggested by the reviewer. To avoid future confusion, we have changed all relevant occurrences of "specificity" into "precision". The following sentence was modified to clarify the definition:

p. 5 line 15

To evaluate the precision (the fraction of called positives that are true positives) of this assay, all cells were collected and analyzed by FACS after image analysis and photo-activation (Fig. 2d and 2e). We calculated precision as the fraction of photo-activated cells (mCherry positive cells) that are true positives (mIFP-mCherry double positive cells) (Fig. 2f).

Measuring recall is complicated because the microscope is unable to visit all locations in the imaging plate, hence recall will depend on the fraction of cells actually “seen” by the microscope. For the screening strategy employed in the nuclear size screen, recall is not as important as precision, since lower recall rates are compensated for by screening larger cell numbers. We therefore did not attempt to measure recall directly.

Page 6 - Related to the above point, the authors state "These results indicate the assay yields reliable hit identification regardless of the percentage of hits in the library." This statement seems too strong given that the authors looked at specificity experimentally with a mixture of ~1% mIFP positive cells. In fact, hits might be much less than 1% of the total population of cells, and specificity would certainly fall from the 80% measured at 1% of the total population. The authors should do a bit more to fairly discuss their ability to find rare hits.

We agree with the reviewer and have changed the following description:

p. 5 line 20

The precision varied with the initial percentage of mIFP positive cells and ranged from 80% to ~100% (initial percentage of mIFP positive cells ranging between 2.3% and 43.7%) (Fig. 2f). Precision is expected to fall below 80% with initial percentage of mIFP positive cells less than 2.3%. However, these results indicate that optical enrichment can be used to identify hits with high precision even at relatively low hit rates.

Pages 6/7 - The authors perform a validation experiment using two different sgRNA libraries, infecting mIFP- and mIFP+ cells separately. Then, they demix these populations via optical enrichment, sequence and compute a phenotype score for sgRNAs or groups of sgRNAs. The way the experiment is described and visualized is extremely confusing. If I understood correctly (and I am not sure that I did), the bottom right panel of Figure 2b shows that if sgRNAs are (randomly?) paired AND two replicates are combined then optical enrichment nearly perfectly separates all (combined, paired) sgRNAs in the two libraries. The authors should rewrite this section, especially clarifying what is meant by "1 sgRNA/group and 2 sgRNA/group," and consider changing Figure 2b (perhaps just show the lower right panel?).

We apologize for our confusing description. To avoid the confusion, we rewrote the paragraph describing the experiment and added a schematic (Fig. 3a) to better describe this experiment. We also simplified the result by just presenting the lower right panel of original Fig. 2b (current Fig. 3b) and moved the other data into supplementary figures (Fig. S2).

p. 6 line 4

mIFP negative cells and mIFP positive cells were separately infected with two different CRISPRa sgRNA libraries (6100 sgRNAs for mIFP negative cells; 860 sgRNAs for mIFP positive cells) at a low multiplicity of infection (MOI) to guarantee a single sgRNA per cell. Note that in these experiments, the sgRNAs only function as barcodes to be read out by sequencing, but do not cause phenotypic changes as the cells do not express corresponding CRISPR reagents. These two populations were then mixed at a ratio of 9:1 mIFP negative cells: mIFP positive cells. We again used mIFP expression as our phenotype of interest (outlined in Fig. 3a). Two biological replicates were performed and at least 200-fold coverage of each sgRNA library was guaranteed throughout the screen, including library infection, puromycin selection, imaging/photo-activation and FACS.

Page 8 - Related to Supplementary Figure 3, why are there not clear BFP+ and BFP- populations but instead one continuous population? How was the gating determined (e.g. how was the boundary between red and gray picked)? Here, and generally, flow plots and histograms of flow plots should indicate the number of cells. If replicates were performed, they should be included.

We have clarified our description. There are no clear BFP+ and BFP- populations but instead one continuous population due to the background expression of BFP from the dCas9 construct: dCas9-KRAB-BFP (which is now clearly indicated in the manuscript). On top of the dCas9-KRAB-BFP, another BFP is encoded on the sgRNA construct, which leads to a higher BFP expression level.

There was no gating in the experiment, the grey dots in the figure represents wild type cells without viral transduction while the red dots (partially covered by the grey dots) were cells infected with the two negative control sgRNAs. We mistakenly wrote the legend of original Fig. S3 (current Fig. S3a) that these were FACS data; however, the data were acquired by imaging. We apologize for the confusion and thank the reviewer for detecting the issue. We completely rewrote the legend to Fig. S3a (original Fig. S3) to clarify.

We now include the number of cells analyzed and the number of replicates for the other flow plots and histograms in the manuscript.

Page 8 - "Nuclear sizes...". The authors should say in the main text what size metric was used.

To address the reviewer's point, we have included the following sentence:

p. 8 line 23

We defined nuclear size as the 2D area in square microns measured by H2B-mGFP using an epifluorescence microscope, as determined by automated image analysis (Fig. 4a and Supplementary file 2).

Page 9 - I am a little confused about the statistical analysis of the screen. In Supplementary File 1, the authors state that p-values were "calculated based on comparison between the distribution of all the phenotypic scores of sgRNAs targeting to the gene/assigning in the group and the one of negative control sgRNAs in the libraries." I presume this means that all phenotypic scores (across replicates) of all sgRNAs targeting each gene were included in a Mann Whitney U test with a single randomized set of phenotypic scores. If that's right, it seems like an odd way to get p-values. Better would be a randomization test, where a null distribution of phenotypic scores for each gene is built by randomizing sgRNA-level scores many times. Then the actual phenotypic score is compared to the randomized null distribution, yielding a p-value. In any case, the authors must clarify what they did in the main text and Supplementary File 1.

Page 9 - It does not appear that the p-values presented in Figure 3c have been adjusted for multiple hypothesis testing. This should be done.

Page 9 - "A value of the top 0.1 percentile of control groups was used as a cutoff for hits." Why? This seems arbitrary. It seems like appropriate false-discovery rate control would enable a more rigorous method for choosing a cutoff.

Page 9 - The same comments regarding analysis and scoring of the optical enrichment screen applies to the FSC and GFP screens.

We clarified the description of the statistical analysis of the screen (see new/changed text below). Mann-Whitney p-values for the two replicates were calculated independently. The Mann-Whitney U test was not performed against a randomized set of phenotypic scores, but using the phenotypic scores of the 22 control non-targeting sgRNAs that were part of the library. Because there are only 22 control sgRNAs (adding more control sgRNAs would increase the size of the library, and reduce the number of genes that can be screened within a given amount of time), the statistical significance of testing genes against these controls is not expected to be very high, and using direct approaches such as multiple hypothesis testing are not expected to yield hits. Instead, we calculated a score combining the severity (phenotypic score) and the trustworthiness (Mann-Whitney p value) of the phenotype (a method previously developed in the Weissman lab at UCSF: <https://github.com/mhorlbeck/ScreenProcessing24>). We thank the reviewer for suggesting using false discovery rate control as a better method for choosing a cutoff. We modified our original analysis and now determine the threshold of our score based on a calculated empirical false discovery rate (eFDR). We used this approach to maximize the number of true hits and relied on a repeat of the screen and follow-up testing of hits to narrow down true hits. We added the following part in the method section and added an analysis example to the supplementary files (Supplementary file 9)."

p. 22

Bioinformatic analysis of the screen

Analysis was based on the ScreenProcessing pipeline developed in the Weissman lab (<https://github.com/mhorlbeck/ScreenProcessing>) (Horlbeck, M. A.; Gilbert, L. A.; Villalta, J. E.; Adamson, B.; Pak, R. A.; Chen, Y.; Fields, A. P.; Park, C. Y.; Corn, J. E.; Kampmann, M.; Weissman, J. S. 2016). The phenotypic score (ϵ) of each sgRNA was quantified as previously defined (Kampmann, M.; Bassik, M. C.; Weissman, J. S. 2013) (Supplementary file 9). For the mIFP proof-of-principle screen, phenotypic score of each group was the average score of two sgRNAs assigned to the group and averaged between two replicates except otherwise described. For the nuclear size screen, FSC screen and H2B-mGFP screen, genes were scored based on the average phenotypic scores of the sgRNAs targeting them. For the nuclear size screen, phenotypic scores were further averaged between 4 runs for each replicate. For the nuclear size screen, FSC screen and H2B-mGFP screen, sgRNAs were first clustered by transcription start site (TSS) and scored by the Mann-Whitney U test against 22 non-targeting control sgRNAs included in the library. Since only 22 control sgRNAs were included, significance of hits was assessed by comparison with simulated negative controls that were generated by random assignment of all sgRNAs in the library and phenotypic scores of these simulated negative controls were scored in the same way as phenotypic scores for genes. A score η that includes the phenotypic score and its significance was calculated for each gene and simulated negative control. The optimal cut-off for score η was determined by calculating an empirical false discovery rate (eFDR) at multiple values of η as the number of simulated negative controls with score η higher than the cut-off (false positives) divided by the sum of genes and simulated negative controls with score η higher than the cut-off (all positives). The cut-off score η resulting in an eFDR of 0.1% was used to call hits for further analysis (Supplementary file 9). An example analysis is described in detail in Supplementary file 9 and raw counts and phenotypic scores for all four screens are listed in Supplementary file 10 and 11.

Page 9 - "These data suggest that a direct measurement utilizing a microscope can provide significant improvement in hit yield even for phenotypes that could be indirectly screened with other approaches." I think this conclusion is too strong. It rests on the assumption that the FSC/GFP phenotypes should have the same set of hits as the microscope phenotype (larger nuclear area). This may not be the case. For example, genes whose inactivation increases GFP expression would be hits in the former, but not latter case. The authors should moderate this statement.

We agree with the reviewer and have changed the sentence into:

p. 10 line 17

These data suggest that a direct measurement utilizing a microscope can provide different information and reveal hits that are inaccessible using other screening approaches.

Page 11 - "This is significantly faster than the in situ methods." The authors should provide a citation and an actual comparison to the speed of in situ methods.

We agree with the reviewer and have modified the sentence with a citation:

p. 12 line 20

This is significantly faster than in situ methods which process millions of cells over a period of a few days(Feldman, D.; Singh, A.; Schmid-Burgk, J. L.; Carlson, R. J.; Mezger, A.; Garrity, A. J.; Zhang, F.; Blainey, P. C. 2019).

Page 12 - I think the authors could say a bit more about the possibility of low hit rate screens. How low do they think it is feasible to go? What hit rates are expected based on existing arrayed optical screens?

We have added more description in the discussion section:

p. 13 the second paragraph

Optical enrichment screening also is possible for phenotypic screens with relatively low hit rates (defined as the fraction of all genes screened that are true hits). The ability to detect hits at low hit rates in our method depends on multiple factors, including: 1) the penetrance of the phenotype; 2) cellular fitness effect of the phenotype; 3) detection and photo-activation accuracy of the phenotype; 4) limitations imposed by FACS recovery and sequencing sample preparations of low cell numbers. The first three factors vary with the phenotype of interest. We optimized the genomic DNA preparation protocol (Methods), and are now able to process sequencing samples from a few thousand cells, enabling screens of low hit rate phenotypes. In our nuclear size screen, more than 1.5 millions cells were analyzed during each run with 2000-4000 cells recovered after FACS sorting. The hit rate of this screen was 2.76%, similar to optical CRISPR screens performed in an arrayed format(de Groot, R.; Luthi, J.; Lindsay, H.; Holtackers, R.; Pelkmans, L. 2018), *demonstrating the possibility to apply our approach to investigate phenotypes with low hit rates.*

Page 14 - It is weird that the discussion includes a fairly important couple of paragraphs that seem to belong in the results (e.g. the text surrounding Figure 4b and c). Obviously, I don't want to prescribe stylistic changes, but I suggest the authors consider moving this description of the experiments/analyses to the results.

The relevant description has been moved to the results.

Page 14 - The authors validate their hits individually, and observe that expression of hit sgRNAs does increase nuclear size in some cells. But, many/most cells remain control-like in these validation experiments. The authors should comment on why this is the case (e.g. inefficient knockdown, cell cycle effects, etc).

To address this point, we have added the following sentences in legend of Fig. 5:

The cell population is heterogeneous due to inefficient knockdown, incomplete puromycin selection, and penetrance of the phenotype. A BFP was expressed from the same sgRNA construct. Only cells with high BFP intensity, indicating successfully sgRNA transduction, were included for data analysis as described in Methods.

Page 14 - It would be nice to formally compare the control and sgRNA distributions in each panel of 4a and Supplementary Figure 5 (e.g. with a Kolmogorov-Smirnov test, etc). That would allow a more precise statement to be substituted for "14 out of 15 hits (the exception was TACC3) were confirmed to be real hits, with cells exhibiting larger nuclei after knock down (Fig. 4a and Fig. S5)," which is not quantitative.

We applied the Kolmogorov-Smirnov test and the corresponding sentence was changed into:

p. 10 last line

14 out of 15 hits were confirmed to be real hits (Kolmogorov-Smirnov test two tailed p-value < 0.01; the exception was TACC3; Fig. 5a and Fig. S5).

Figure 2a - I am not sure it is necessary to show the entire workflow again. The first and possibly last panels are the informative ones here.

Figure 3a - Same comment as above - these workflow panels take up a lot of real estate and I suggest simplifying them if possible.

The figures were simplified to just show the example images.

Figure 3c - At least on my PDF/screen, the "scrambled control" points appear very light gray and are impossible to find. They should be an easier to spot color.

We agree with the reviewer and changed the color.

Figure 4b - "Most cells developed a larger cellular size and higher H2B-mGFP level after knock down." I think it would be more accurate to say that the median cell size/GFP level increased, or that some cells developed larger sizes/median GFP levels.

We agree with the reviewer's point; "most" has been changed to "some".

Figure 4c - I don't understand "Normalized FITC/nuclear size." Do the bars show the mean/median of a population (if so, why not show a dot plot or box plot or violin plot)? Also, what is FITC (I presume it's GFP levels)?

Figure 4c - "Most cells maintained a constant ratio between nuclear size and DNA content..." I'm not sure where DNA content came from. Are the authors assuming that their H2B-mGFP is a proxy for DNA content? Or was some other measurement made? If the former, is there a citable reason why this is a good assumption?

The bars represent the ratio of the median level of H2B-mGFP intensity (the axis is now labeled with "GFP" rather than "FITC", the colloquial name for the channel used on the FACS machine) measured by FACS and the median nuclear size of the same population of cells measured by microscopy. We plan to perform additional experiments to measure DNA content using a DNA dye in the same cell by microscopy so that we

will be able to correlate these on a cell by cell basis. Data will be added before final publication.

Reviewer #2 (Significance (Required)):

I don't generally comment on significance in reviews. Since ReviewCommons is specifically asking, I'll say that this manuscript describes optical enrichment, a method that is an extension of previous work and is substantially similar to a previously published method, Visual Cell Sorting. However, given the timing, it is obvious that these authors have been working independently on optical enrichment. Since the application is distinct, and optical enrichment incorporates some nice features like software to make it easier to execute, it is clearly of independent value.

Reviewer #3 (Evidence, reproducibility and clarity (Required)):

This study reports a rapid and high-throughput CRISPR-based phenotypic screen approach consisting of selecting cells with phenotypes of interest, label them by photo-conversion and isolating them by FACS. The idea of the method is interesting (has been around) in principle. The key advantage is that is relatively simple, accessible to many groups as it does not require robotics. However, the manuscript is so badly written and hard to follow, that it makes it difficult to judge the technology, to really understand how the experiments were done and whether the results are interpreted correctly. Strictly speaking, it is unclear whether and how good scientific practices GSP have been followed, as the description of the experiments is sometimes lacking totally. Consequently, it is impossible to seriously evaluate this study and judge whether the technology described is really promising. It is probably less sensitive than arrayed screens, in all likelihood can miss hits that affect growth, cannot capture as many phenotypic classes as one would like from high-content screens and the computational and experimental workflow is more complicated. It is puzzling that the authors don't even compare the results with arrayed screens which are of course the current gold-standard.

We do not in any way claim that the presented method replaces arrayed screens. However, most current sgRNA libraries are pooled libraries, and the few available arrayed sgRNA libraries are expensive and difficult to maintain, hence our methods to screen pooled sgRNA libraries are timely and useful. Comparisons with arrayed screens are unwarranted as no claims are made with respect to arrayed screens.

We have clarified the manuscript in many places, and hope it is now readable and better understandable by more readers with diverse backgrounds.

****Specific points:****

The specificity test (Fig 1) does not make sense how it is described. If the authors spike a certain percentage of cells that can be photoconverted, when analysing the outcome, there will be three classes: mIFP positive, mIFP/mCherry positive and negative. How

can they calculate specificity if they do not know whether they converted all mIFP cells? Also the formula used is questionable or is her an error? Furthermore, it is totally unclear how many cells were used and how they were scanned. If they took 90 negative cells and 10 mIFP cells, getting them all back is easy. If they start with 10e9 cells, the specificity should be quantified. Furthermore, the phenotype they pick is an easy and convenient one. Much more challenging is to apply it on a multi-parametric phenotype. Again, this is now the gold standard.

We used the term specificity inadvertently and should have used precision, as also pointed out by Referee 2. This has been corrected in the current manuscript. We picked the mIFP phenotype as this was a proof of principle screen to clarify the performance of our screening approach and needed a phenotype that can be measured both by microscopy and FACS. We demonstrate that multi-parametric read-outs are possible, but do not think that the first demonstration of new technology needs such an application.

In their first sgRNA assay, it is not possible to have a clear idea of what groups they are talking about. Do they mean they get phenotypic signatures which they group? How? They need to describe what they do. Here, only ~3500 genes are scanned (the 6843 is both populations and you only select from the mIFP neg population) and it took them 8hrs. This means for the genome it would require ~60h which is indeed fast. However, this experiment is not clearly described. They cannot select the negative population since there is no fluorescent marker (except false positive which are around 1.7%). So I assume they just randomly pick cells (they should really explain much better what they do!). Why go through the hassle? If these sequences are supposed to be a negative population, just pick them in the computer. Also, they cannot calculate an enrichment compared to the negative population, since two different libraries were infected. Again, I can't follow.

We improved the description of this experiment. To clarify, we used mIFP in a proof of concept screen to validate whether sgRNAs infecting mIFP positive cells can be distinguished from those infecting mIFP negative cells. No phenotypic signature other than the mIFP signal is used (as described in the text). As customary in pooled screens, a primary comparison was made between the positive (optically selected) cells and the complete population. To improve the clarity of this screen, we further described the concept of pooled sgRNA screens, which may have made this section harder to follow.

I find their results about calculating scores based only on true negatives surprising. The average phenotypic score is improved from 3 to 5, which is enormous. This suggests that the phenotypes induced in the mIFP population are extremely common. These results are hard to interpret given the poor description of the experiment. It is possible that it is the same dataset as in 1, but in that case, the false negatives must be rare since the negatives can be selected by absence of both mCherry and mIFP.

There are no phenotypes induced in the mIFP population (as now explicitly explained in the text). The mIFP population is isolated using optical enrichment, and we test our

ability to discriminate the sgRNAs present in the enriched population. It is unsurprising that comparing to the negatively selected population (which is not possible in most other pooled screens) is significantly better than comparing against the total population (as customary in pooled screens).

In the nuclear size screen, 6000 sgRNAs were screened. To array so many sequences would require 20 plates. They required ~40h for imaging one replicate. This is slow, imagine the time with a 60x lens.

There are no arrayed screens performed in our study.

Reviewer #3 (Significance (Required)):

Overall, there is no sufficient evidence in this manuscript to convince this reviewer that this method is valid and truly powerful. I cannot support publication in its present form.

October 21, 2020

RE: JCB Manuscript #202008158T

Dr. Ronald D Vale
Janelia Research Campus
19700 Helix Drive
Asburn, VA 20147

Dear Ron:

Thank you for submitting your revised manuscript entitled "High-content Imaging-based Pooled CRISPR Screens in Mammalian Cells". Your paper has been seen again by the original reviewers, all of whom now recommend acceptance provided that the promised qPCR data is included in the final manuscript. Thus, we would be happy to publish your paper in JCB pending these final revisions and others necessary to meet our formatting guidelines (see details below).

****As mentioned above, please be sure to add the qPCR data to the final revision and clearly mark any new text and/or references to the new data in the main manuscript. Final acceptance of the paper is dependent on the inclusion of this data.****

A. MANUSCRIPT ORGANIZATION AND FORMATTING:

Full guidelines are available on our Instructions for Authors page, <https://jcb.rupress.org/submission-guidelines#revised>. ****Submission of a paper that does not conform to JCB guidelines will delay the acceptance of your manuscript.****

1) Text limits: Character count for Tools is < 40,000, not including spaces. Count includes title page, abstract, introduction, results, discussion, and acknowledgments. Count does not include materials and methods, figure legends, references, tables, or supplemental legends. You are currently below this limit but please bear it in mind when revising.

2) Figures limits: Tools may have up to 10 main text figures.

3) Figure formatting: Scale bars must be present on all microscopy images, including inset magnifications. Molecular weight or nucleic acid size markers must be included on all gel electrophoresis.

4) Statistical analysis: Error bars on graphic representations of numerical data must be clearly described in the figure legend. The number of independent data points (n) represented in a graph must be indicated in the legend. Statistical methods should be explained in full in the materials and methods. For figures presenting pooled data the statistical measure should be defined in the figure legends. Please also be sure to indicate the statistical tests used in each of your experiments (both in the figure legend itself and in a separate methods section) as well as the parameters of the test

(for example, if you ran a t-test, please indicate if it was one- or two-sided, etc.). Also, if you used parametric tests, please indicate if the data distribution was tested for normality (and if so, how). If not, you must state something to the effect that "Data distribution was assumed to be normal but this was not formally tested."

5) Materials and methods: Should be comprehensive and not simply reference a previous publication for details on how an experiment was performed. Please provide full descriptions (at least in brief) in the text for readers who may not have access to referenced manuscripts. The text should not refer to methods "...as previously described."

6) Please be sure to provide the sequences for all of your primers/oligos and RNAi constructs in the materials and methods. You must also indicate in the methods the source, species, and catalog numbers (where appropriate) for all of your antibodies.

7) Microscope image acquisition: The following information must be provided about the acquisition and processing of images:

- a. Make and model of microscope
- b. Type, magnification, and numerical aperture of the objective lenses
- c. Temperature
- d. imaging medium
- e. Fluorochromes
- f. Camera make and model
- g. Acquisition software
- h. Any software used for image processing subsequent to data acquisition. Please include details and types of operations involved (e.g., type of deconvolution, 3D reconstitutions, surface or volume rendering, gamma adjustments, etc.).

8) References: There is no limit to the number of references cited in a manuscript. References should be cited parenthetically in the text by author and year of publication. Abbreviate the names of journals according to PubMed.

9) Supplemental materials: There are usually strict limits on the allowable amount of supplemental data. Articles/Tools may have up to 5 supplemental figures. You currently have 6 such figures. We should be able to allow you the extra space this time but please try not to add to this total. Please also note that tables, like figures, should be provided as individual, editable files. A summary of all supplemental material should appear at the end of the Materials and methods section.

10) eTOC summary: A ~40-50 word summary that describes the context and significance of the findings for a general readership should be included on the title page. The statement should be written in the present tense and refer to the work in the third person. It should begin with "First author name(s) et al..." to match our preferred style.

11) Conflict of interest statement: JCB requires inclusion of a statement in the acknowledgements regarding competing financial interests. If no competing financial interests exist, please include the following statement: "The authors declare no competing financial interests." If competing interests are declared, please follow your statement of these competing interests with the following statement: "The authors declare no further competing financial interests."

12) A separate author contribution section is required following the Acknowledgments in all research manuscripts. All authors should be mentioned and designated by their first and middle

initials and full surnames. We encourage use of the CRediT nomenclature (<https://casrai.org/credit/>).

13) ORCID IDs: ORCID IDs are unique identifiers allowing researchers to create a record of their various scholarly contributions in a single place. At resubmission of your final files, please consider providing an ORCID ID for as many contributing authors as possible.

B. FINAL FILES:

-- High-resolution figure and video files: See our detailed guidelines for preparing your production-ready images, <https://jcb.rupress.org/fig-vid-guidelines>.

****It is JCB policy that if requested, original data images must be made available to the editors. Failure to provide original images upon request will result in unavoidable delays in publication. Please ensure that you have access to all original data images prior to final submission.****

****The license to publish form must be signed before your manuscript can be sent to production. A link to the electronic license to publish form will be sent to the corresponding author only. Please take a moment to check your funder requirements before choosing the appropriate license.****

Thank you for this interesting contribution, we look forward to publishing your paper in Journal of Cell Biology.

Sincerely,

Jodi Nunnari, PhD
Editor-in-Chief
The Journal of Cell Biology

Tim Spencer, PhD
Executive Editor

Reviewer #1 (Comments to the Authors (Required)):

The authors have mostly addressed my concerns. While think this manuscript is appropriate for JCB, I would highly recommend to include the qPCR data before accepting this manuscript for publication.

Reviewer #2 (Comments to the Authors (Required)):

The authors have addressed my concerns and, pending the inclusion of the data promised in the rebuttal, I recommend publication.

Reviewer #3 (Comments to the Authors (Required)):

The authors have improved the readability of the manuscript substantially and the logic and intention of the experiments are much clearer to me. The authors demonstrate convincingly the validity of optical selection in pooled screens and it is important for researchers to be aware of alternative methods for screening CRISPR libraries. I therefore recommend the publication of the manuscript in its current form.

Ronald D. Vale, Ph.D.
Investigator, Howard Hughes Medical Institute
Executive Director, Janelia Research Campus
Professor, Dept. of Cellular and Molecular Pharmacology, UCSF
600 16th Street, San Francisco, CA 94143

Telephone: (415) 476-6380
Fax: (415) 476-5233
Email: valer@janelia.hhmi.org

Nov. 14, 2020

Drs. Jodi Nunnari and Tim Spencer
The Journal of Cell Biology

Dear Jodi and Tim,

On behalf of my colleagues, we are pleased to submit our final manuscript entitled “High-Content Imaging-Based Pooled CRISPR Screens in Mammalian Cells” as a “Tool” article for the Journal of Biological Sciences. Our paper was reviewed and recommended for acceptance by three referees through Review Commons with promised RT-qPCR data. The data is now included in Figure S5 and Supplementary file 12 with the description of the experiment “11 out of 15 genes showed >75% knockdown, as revealed by RT-qPCR with most genes demonstrating almost complete knock down (Fig. S5).” highlighted in blue in our main text.

Sincerely yours,

Ron Vale
Professor of Cellular and Molecular Pharmacology, UCSF
Executive Director, Janelia Research Campus